# Graphons of Line Graphs

## Abstract

We consider the problem of estimating graph limits, known as graphons, from observations of sequences of sparse finite graphs. In this paper we show a simple method that can shed light on a subset of sparse graphs. The method involves mapping the original graphs to their *line graphs*. We show that graphs satisfying a particular property are sparse, but give rise to dense line graphs. This property, the *square-degree property*, enables us to apply results on graph limits of dense graphs to derive convergence. In particular, star graphs satisfy the square-degree property resulting in dense line graphs and non-zero graphons of line graphs. We demonstrate empirically that we can distinguish different numbers of stars (which are sparse) by the graphons of their corresponding line graphs. Whereas in the original graphs, the different number of stars all converge to the zero graphon due to sparsity. Similarly, superlinear preferential attachment graphs give rise to dense line graphs almost surely. In contrast, dense graphs, including Erdős–Rényi graphs make the line graphs sparse, resulting in the zero graphon.

## 1 Introduction

A graphon is the limit of a converging graph sequence. Graphons of dense graphs are useful as they can act as a blueprint and generate graphs of arbitrary size with similar properties. But for sparse graphs this is not the case. Sparse graphs converge to the zero graphon, making the generated graphs empty or edgeless. Thus, the classical graphon definition fails for sparse graphs. Several methods have been proposed to overcome this limitation and to understand sparse graphs more deeply. However, the nature of sparse graphs makes these methods mathematically complex. Graphons are useful in machine learning as a prior distribution on graphs. Graphons provide an interesting connection between combinatorial, probabilistic, and analytical problems, leading to many new approaches for graph modelling.

The obvious use of graphons is to predict a network and its properties at a future time point when the network is large (Chayes, 2016). The fact that graphons are compact objects with the ability to generate arbitrarily large networks is an attractive feature. It is also studied in the context of exchangeable arrays (Orbanz & Roy, 2015). In addition to network prediction, graphons are used in a myriad ways including in tranfer learning neural networks (Ruiz et al., 2020), graph embeddings (Davison & Austern, 2023) and motif sampling (Lyu et al., 2023). They are also of interest to problems in extremal graph theory, the study of large graphs and random matrix theory. Graphons have had wide application in statistical physics and network theory.

The theory of graphons of dense graphs is well developed, and is based on the Aldous-Hoover theorem. For a graphon to exist the sequence of graphs need to converge in homomorphism density, which can be thought of as subgraph density. However, a limitation of such graphons is that they produce dense graphs when the graphon is non-zero. If the graphon is zero everywhere, then it is of little use as it can only produce an empty graph. Thus, sparse graphs cannot be modelled using this approach. The classical constructions prevent models where the number of edges grow sub-quadratically with respect to the number of nodes. Previous alternative approaches for sparse graphons include constructions using Kallenberg exchangeability (Caron & Fox, 2017), stretched graphons (Borgs et al., 2018) and graphexes (Borgs et al., 2021).

In this paper, we propose a new way to model sparse graphons by modeling the graphon of the corresponding line graph. Line graphs map edges to vertices and connects edges when edges in the original graph share

a vertex. For a graph $G_n$ with $n$ nodes, a line graph $H_m := L(G_n)$ is a graph where each of the $m$ edges of the original graph $G_n$ is a node of $H_m$. Many properties of the original graph $G_n$ have a corresponding property in the line graph $H_m$. In contrast to previous approaches to graphons of sparse graphs that required complex mathematical machinery, our approach builds on the results of graphons on dense graphs directly. We discover that if graphs $G_n$ have the property that the sum of the squares of the node degrees is greater than the square of the number of edges, then the corresponding line graphs $H_m$ are dense. This relationship between $G_n$ and $H_m$ may be of independent interest. We show that sparse graphs $G_n$ that satisfy the so called "square-degree property" have line graphs $H_m$ that result in non-zero graphons.

We provide some background in Section 2, and present our discovery connecting graphs $G_n$ with their line graphs $H_m$ in Section 3. We show that graphs $G_n$ that satisfy the square-degree property have convergent edge densities and homomorphism densities. We derive the graphons for disjoint star graphs in Section 4 and illustrate the empirical behaviour of estimation on sparse graphs in Section 4.4. We derive graphons of line graphs for preferential attachment and Erdos-Renyi graphs in Section 5.

**Contributions of this paper**

- We propose a property of sparse graphs, the square-degree property (Definition 3.3) which allows us to find sparse graphs whose line graphs are dense. In particular, sparse graphs with square-degree property have dense line graphs, and under certain conditions have line graph limits (Section 3.4).

- We prove that for disjoint star graphs, the corresponding line graphs are dense and hence have graph limits (Section 4). Furthermore, we show that certain preferential attachment graphs have dense line graphs that converge to non-zero graphons under certain conditions (Section 5.1).

- We illustrate with empirical graphons the utility of line graphs for sparse graphs in Section 4.4.

## 2 Notation and Preliminaries

A simple graph is a graph without loops or multiple edges between the same nodes. We only consider simple graphs and sequences of simple graphs in this paper.

### 2.1 Line graphs

Let $G$ denote a graph. If $G$ has at least one edge, then its line graph is the graph whose vertices are the edges of $G$, with two of these vertices being adjacent if the corresponding edges are adjacent in $G$ (Beineke & Bagga, 2021). Figure 1 shows an example of a graph and its line graph. The edges in the graph on the left are mapped to the vertices in the line graph (on the right) as can be seen from the numbers.

We denote the line graph operation by $L$, i.e., for a graph $G$ we denote its line graph by $H := L(G)$. In terms of notation we make a distinction between graphs $G$ and line graphs $H$, i.e., we use the letter $H$, with and without subscripts, to denote line graphs.

Rather than a single graph $G$, we are interested in graph sequences. The exact type of sequences which forms our interest will be made clear by the end of this section. Let $\{G_n\}_{n=1}^{\infty}$ denote a graph sequence. The index $n$ denotes the number of nodes in $G_n$ and let the number of edges be given by $m$. We denote the line graph of $G_n$ by $H_m := L(G_n)$ as $H_m$ has $m$ nodes.

We use standard graph theory notation to denote specific types of graphs. As customary $K_n$ denotes a complete graph of $n$ nodes, and $K_{s,r}$ denotes a complete bi-partite graph of partition sizes $s$ and $r$, i.e., there are $s$ nodes in one subset completely connected to $r$ nodes in the other subset. When $s = 1$ we get star graphs; $K_{1,n}$ denotes a star with $n + 1$ vertices, where $n$ vertices are connected to the hub vertex.

**Definition 2.1.** *If $G$ is a graph whose line graph is $H$, that is, $L(G) = H$, then $G$ is called the **root** of $H$.*

Whitney (1932) showed that the structure of a graph can be recovered from its line graph with one exception: if the line graph $H$ is $K_3$, a triangle, then the root of $H$ can be either $K_{1,3}$, a star or $K_3$ a triangle. This follows from the following theorem as stated in Harary (1969):

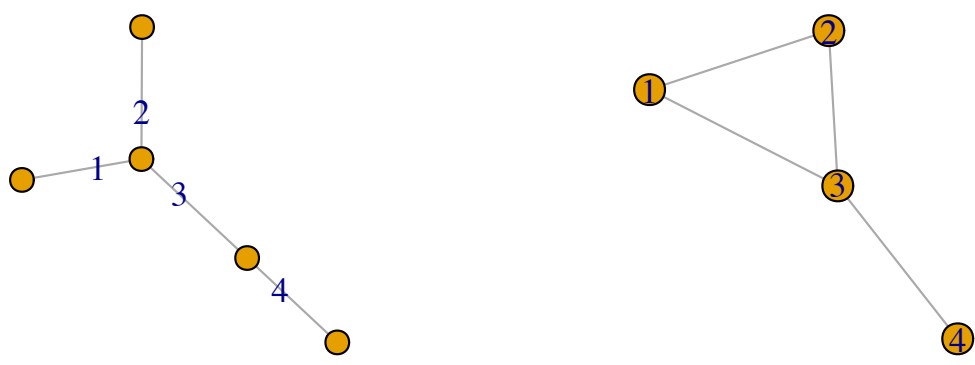

**Figure 1:** *A graph on the left and its line graph on the right.*

**Theorem 2.2** (Whitney1932, Harary 1969)**.** *Let $G$ and $G'$ be connected graphs with isomorphic line graphs. Then $G$ and $G'$ are isomorphic unless one is $K_3$ and the other is $K_{1,3}$.*

By simply creating edges corresponding to vertices in line graph $H$ and connecting them by merging the vertices if there is an edge between the vertices in $H$ we can obtain the the graph $G$, such that $H = L(G)$. Thus, if $H$ is a line graph and it is not $K_3$, then we can talk about $L^{-1}(H)$.

We state some preliminary results on line graphs covered in Chapter 1 of Beineke & Bagga (2021).

**Lemma 2.3.** *Let $G$ be a non-null graph with $n$ vertices and $m$ edges. Let $H = L(G)$. Then*

1. *$H$ has $m$ vertices and $\frac{1}{2}\sum(\deg\ v)^2 - m$ edges*

2. *If $G$ is an $r$-regular graph then $H$ is $2(r-1)$-regular and has $\frac{nr}{2}$ vertices.*

3. *If $G$ is a path $P_n$, then $H$ is also a path of $n-1$ vertices, i.e., $H = P_{n-1}$.*

4. *If $G$ is a non-trivial connected graph, then $H$ is also connected.*

5. *If $G$ is a cycle $C_n$ of $n$ vertices, then $H$ is also a cycle $C_n$ of $n$ vertices.*

6. *If $G$ is a star, i.e., $G = K_{1,n-1}$, then $H$ is a complete graph of $n-1$ vertices, i.e. $H = K_{n-1}$.*

The edge density of a graph $G$ with $n$ nodes and $m$ edges is given by density$(G) = \frac{2m}{n(n-1)}$. Thus, from Lemma 2.3(1) the edge density of $H = L(G)$ is given by

$$\text{density}(H) = \frac{\frac{1}{2}\sum(\deg\ v^2) - m}{\frac{1}{2}m(m-1)}, \tag{1}$$

where deg $v$ denotes the degree distribution of graph $G$ and deg $v^2$ denotes the vector of squared degrees in $G$. We refer to the edge density simply as density.

## 2.2 Graphons

Next we turn our attention to graphons. A graphon is a symmetric, measurable function $W : [0,1]^2 \rightarrow [0,1]$ often used to describe both the limiting properties of graph sequences as well as the graph generation process (Borgs et al., 2011). We define some terms often used in the graphon literature.

**Definition 2.4.** *A **graph homomorphism** from $F$ to $G$ is a map $f : V(F) \to V(G)$ such that if $uv \in E(F)$ then $f(u)f(v) \in E(G)$. (Maps edges to edges.) Let $Hom(F, G)$ be the set of all such homomorphisms and let $\hom(F, G) = |Hom(F, G)|$. Then **homomorphism density** is defined as*

$$t(F, G) = \frac{\hom(F, G)}{|V(G)|^{|V(F)|}} \,.$$

*The number of homomorphisms $\hom(F, G)$ is given by*

$$\hom(F, G) = \sum_{\phi : V(F) \to V(G)} \prod_{uv \in E(F)} \beta_{\phi(u)\phi(v)}(G)$$

*where $\beta_{ij}(G)$ is the weight of edge $ij$ in graph $G$, which equals either 1 or 0 in unweighted graphs. For a graphon $W$, the homomorphism density is defined as*

$$t(F, W) = \int_{[0,1]^{|V(F)|}} \prod_{ij \in E(F)} W(x_i, x_j) \, dx \,.$$

A graph homomorphism is an edge preserving map from one graph to another. The homomorphism density is useful as it is bounded even when the number of homomorphisms $\hom(F, G)$ go to infinity.

**Definition 2.5.** *The **cut norm** of graphon $W$ (Frieze & Kannan, 1999; Borgs et al., 2008) is defined as*

$$\|W\|_\square = \sup_{S,T} \left| \int_{S \times T} W(x, y) \, dx dy \right| ,$$

*where the supremum is taken over all measurable sets $S$ and $T$ of $[0, 1]$.*

**Definition 2.6.** *Given two graphons $W_1$ and $W_2$ the **cut metric** (Borgs et al., 2008) is defined as*

$$\delta_\square(W_1, W_2) = \inf_\varphi \|W_1 - W_2^\varphi\|_\square ,$$

*where the infimum is taken over all measure preserving bijections $\varphi : [0, 1] \to [0, 1]$.*

Let $\mathcal{W}$ denote the space of graphons, i.e., $\mathcal{W} = \{W \in \mathcal{W}\}$. Then, the cut metric is a pseudo-metric in $\mathcal{W}$ because $\delta_\square(W_1, W_2) = 0$ does not imply $W_1 = W_2$, i.e., $\delta_\square(W_1, W_2) \geq 0$ for $W_1 \neq W_2$. However the cut metric $\delta_\square$ is a metric on the quotient space $\tilde{\mathcal{W}} = \mathcal{W}/\sim$ where $f \sim g$ if $f(x, y) = g(\sigma x, \sigma y)$ for some measure preserving $\sigma$.

**Definition 2.7.** *Uniformly pick $x_1, x_2, \dots x_n$ from $[0, 1]$. A **W-random graph** $\mathbb{G}(n, W)$ has the vertex set $1, 2, \dots n$ and vertices $i$ and $j$ are connected with probability $W(x_i, x_j)$.*

We can think of $W$-random graphs as graphs sampled from the graphon $W$. We will use $W$-random graphs in our experiments.

The homomorphism density is used to define graph convergence.

**Definition 2.8** ((Borgs et al., 2008)). *A graph sequence $\{G_n\}_n$ is said to be convergent if $t(F, G_n)$ converges as $n$ goes to infinity for any simple graph $F$.*

Every finite, simple graph $G$ can be represented by a graphon $W_G$, which we call its empirical graphon.

**Definition 2.9.** *Given a graph $G$ with $n$ vertices labeled $\{1, \dots, n\}$, we define its **empirical graphon** $W_G : [0, 1]^2 \to [0, 1]$ as follows: We split the interval $[0, 1]$ into $n$ equal intervals $\{I_1, I_2, \dots, I_n\}$ (first one closed, all others half open) and for $x \in I_i, y \in I_j$ define*

$$W_G(x, y) = \begin{cases} 1 & if \quad ij \in E(G) \\ 0 & otherwise \,, \end{cases}$$

where $E(G)$ denotes the edges of $G$. The empirical graphon replaces the the adjacency matrix with a unit square and the $(i, j)$th entry of the adjacency matrix is replaced with a square of size $(1/n) \times (1/n)$.

The cut metric between graphs $G$ and $G'$ is defined as $\delta_\square(G, G') = \delta_\square(W_G, W_{G'})$. The cut metric between a graph $G$ and a graphon $U$ is defined as $\delta_\square(G, U) = \delta_\square(W_G, U)$.

Borgs et al. (2008) prove the following theorem for convergent graph sequences.

**Theorem 2.10** (Borgs et al. (2008)). *For every convergent sequence $\{G_n\}_n$ of simple graphs there is a graphon $W$ with values in $[0, 1]$ such that $t(F, G_n) \to t(F, W)$ for every simple graph $F$. Moreover for every graphon $W$ with values in $[0, 1]$ there is a convergent sequence of graphs satisfying this relation.*

**Theorem 2.11** (Borgs et al. (2011)). *A sequence of graphs $\{G_n\}_n$ is convergent if and only if it is Cauchy in the $\delta_\square$ distance. The sequence $\{G_n\}_n$ converges to $W$ if and only if $\delta_\square(W_{G_n}, W) \to 0$. Furthermore, if this is the case, and $|V(G_n)| \to \infty$, then there is a way to label the nodes of the graphs $G_n$ such that $\|W_{G_n} - W\|_\square \to 0$.*

### 2.2.1 Line graphs and edge exchangeability

As discussed above edge-exchangeable graphs can exhibit sparsity (Janson, 2018). Here we show the link between line graphs and edge exchangeability.

Figure 2 shows the connection between vertex and edge exchangeability when we map from graphs to line graphs. Graph $G$ is shown on the top left and its line graph $H = L(G)$ is shown on the top right. The graph on the bottom right $H'$ is $H$ with vertices permuted. Let us call the graph on the bottom left $G'$. Following definition 2.1 we can see that $G'$ is the root of $H'$, i.e., $H' = L(G')$. Furthermore, the vertex permutation $\phi$ relabeled the vertices $(1, 2, 3, 4)$ in $H$ to $(2, 3, 4, 1)$ in $H'$. We see the same permutation occurs in edges from $G$ to $G'$, i.e. $G'$ is an edge permuted version of $G$. This is not surprising as line graphs map edges to vertices.

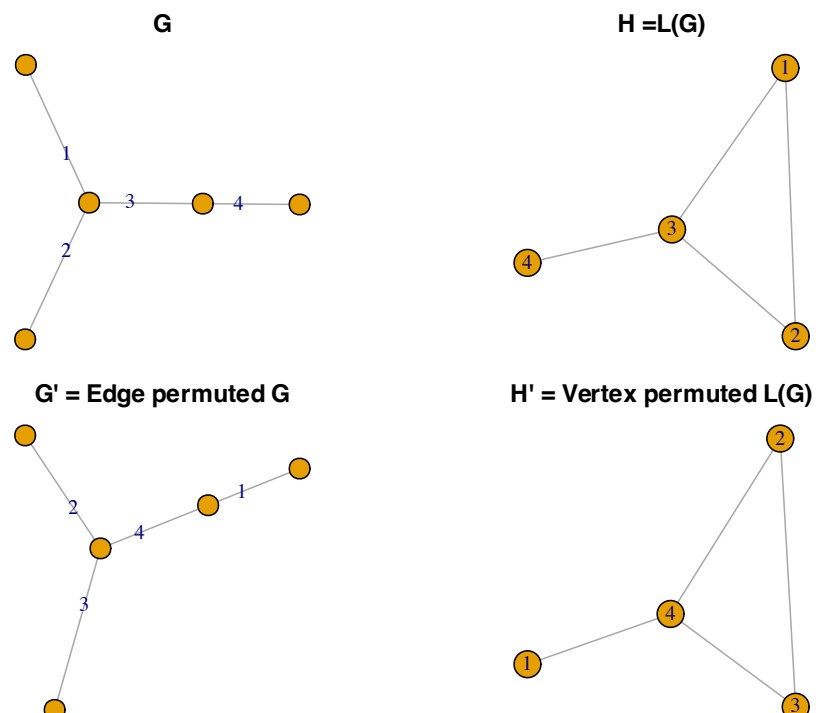

**Figure 2:** *Vertex and edge exchangeability in graphs and line graphs. Graphs $G$ and $H = L(G)$ on the top row. Graph $H'$ is a vertex permuted version of $H$. We see that $H' = L(G')$, where $G'$ is the edge permuted version of $G$.*

### 2.2.2 Edge vs homomorphism density

In this study we mention different types of convergence: convergence with respect to homomorphism density (Definition 2.4), cut metric (Definition 2.6), and edge density (Equation 1). Homomorphism density convergence is subgraph convergence. Suppose $\{G_n\}_n$ converges in homomorphism density, then for any graph $F$ the sequence $\{t(F, G_n)\}_n$ converges. That is, the edge density, triangle density, 4-cycle density and all such densities converge. Convergence in homomorphism density is equivalent to convergence in the cut metric as shown by Borgs et al. (2011). In contrast, edge density convergence is the same as convergence of the single sequence $\{t(\bullet\text{-}\bullet, G_n)\}_n$. As edge density is given by $2|E(G_n)|/n(n-1)$ and $t(\bullet\text{-}\bullet, G_n) = 2|E(G_n)|/n^2$ convergence in one implies convergence in the other. The denominators are different because the edge density excludes the diagonal of the adjacency matrix whereas $\{t(\bullet\text{-}\bullet, G_n)\}_n$ includes it (see Definition 2.4). However, edge density is much weaker and does not give us subgraph convergence.

We use edge density to characterize a bigger space of graph sequences – sequences that do not converge either in the cut metric or in edge density. The use of $\liminf$ in the definition of dense graph sequences (Definition 3.1) means that we do not need convergence of edge densities to call a graph sequence dense.

## 2.3 Related work

### 2.3.1 Graphons of sparse graphs

Caron & Fox (2017) set aside the discrete version of exchangeability and consider its continuous counterpart – Kallenberg exchangeability (Kallenberg, 1990). They consider exchangeable point processes and model graphs using completely random measures. They show that by selecting an appropriate Lévy measure, they can construct sparse or dense graphs. Collaborations led by Borgs and Chayes have resulted in considerable work on sparse graph limits. Borgs et al. (2017) consider sparse graph convergence by introducing a new notion of convergence called LD-convergence, which is based on the theory of large deviations. The large deviations rate function is considered to be the limit object for the sparse graph sequence. In Borgs et al. (2018), they introduce *stretched graphons* as a way to overcome the zero graphon, which is the natural limit of sparse graphs. They consider both the *rescaled graphon* introduced by Bollobás & Riordan (2011) and the stretched graphon as means of representing sparse graph limits. In Borgs et al. (2019a) they develop the theory of $L^p$ graphons, which provides convergence for sparse graphs with the flexibility to account for power laws. Borgs et al. (2019b) and Borgs et al. (2021) consider graphexes – a triple including a positive number, a positive integrable function and a graphon – as a framework for modelling sparse graphs.

Edge-exchangeability is another avenue used to model sparse graphs. Instead of considering exchangeability of vertices, edges are labelled and their permutations are considered. Crane & Dempsey (2018; 2019) introduce edge-exchangeable network models and show that these models allow for sparse structure and power-law degree distributions. Cai et al. (2016) consider projective, edge-exchangeable graphs and obtain sparsity results for all Poisson point process-based graph frequency models. Janson (2018) extends the model put forward by Crane & Dempsey (2018) and investigate different types of graphs that can be generated by this model. He shows that graphs ranging from dense to very sparse graphs can be generated by using the Poisson construction.

### 2.3.2 Other graphon applications

Possibly due to its rich mathematical context, graphons are used in many topics in machine learning. For example, it is desirable for a machine learning model to be transferable. Ruiz et al. (2020) propose graphon neural networks as the limit of graph neural networks (GNNs) with the aim of producing transferable GNNs. They show that GNNs are transferable between deterministic graphs obtained from the same graphon. Graphons and the associated theory is used to bolster theoretical aspects of other topics. Levie (2023) propose a graph signal similarity measure for message passing neural networks based on the graphon cut distance. Hence they extend the cut distance to graph signals. Graph embeddings are used for a myriad of downstream tasks such as node classification, clustering and link prediction. Davison & Austern (2023) investigate theoretical aspects of graph embeddings and show that embedding methods implicitly fit graphon models. Under the assumption the graph is exchangeable, they describe the limiting distribution of embeddings

learned via subsampling the network. Graph homomorphisms are closely connected to graphons. Lyu et al. (2023) introduce motif sampling, which essentially sampling graph homomorphisms uniformly at random. They propose two MCMC algorithms for sampling random graph homomorphisms.

# 3 Sparse graphs with dense line graphs

In this section, we show that there are sparse graphs whose line graphs are dense. In particular we show in Theorem 3.6 that sparse graphs with square-degree property (Definition 3.3) have corresponding line graphs that are dense, and vice versa. We show in Section 3.4 that under certain conditions, the corresponding line graphs converge with respect to the homomorphism density, leading to graphons of line graphs. Therefore, this enables us to define a novel approach to defining graph limits for sparse graphs by their associated line graphs. Recall we denote graph sequences as $\{G_n\}_n$ and the corresponding line graph sequence as $\{H_m\}_m$. If the sequences converge, then we consistently use $W$ and $U$ for graphons corresponding to $\{G_n\}_n$ and $\{H_m\}_m$ respectively. We defer many of the proofs of lemmas and theorems to Appendix A.

## 3.1 Graph sequences

**Definition 3.1** (**Dense graph sequences**). *A sequence of graphs $\{G_n\}_n$ is dense if the number of edges $m$ grow quadratically with the number of nodes $n$, i.e.,*

$$\liminf_{n \to \infty} \frac{m}{n^2} = c > 0 \, .$$

*We denote the set of all dense graph sequences by $D$.*

**Definition 3.2** (**Sparse graph sequences**). *A sequence of graphs $\{G_n\}_n$ is sparse if the number of edges $m$ grow sub-quadratically with the number of nodes $n$, i.e.,*

$$\lim_{n \to \infty} \frac{m}{n^2} = 0 \, .$$

*We denote the set of all sparse graph sequences by $S$.*

For dense graph sequences, the density is bounded from below by a non-zero constant, whereas for sparse graph sequences it goes to zero. The density or $m/n^2$ of a sequence of dense graphs $\{G_n\}_n$ does not necessarily converge; the $\liminf$ is strictly positive, i.e., any converging subsequence has strictly positive density as $n \to \infty$. In contrast, the density or $m/n^2$ of sparse graphs converge to zero, i.e., the limit is equal to zero, not just the $\liminf$. The set of dense graph sequences $D$ and the set of sparse graph sequences $S$ is non-intersecting. Furthermore, the complement of the union of $D$ and $S$, $\overline{D \cup S}$ is non-empty. It contains graph sequences $\{G_n\}_n$ such that $\liminf_{n\to\infty} m/n^2 = 0 \neq \limsup_{n\to\infty} m/n^2$, i.e, it is a mixture of dense and sparse graph sequences with the density of different subsequences converging to different limits with some converging to zero.

Next we define a property of a graph sequence that we call the *square-degree property* .

**Definition 3.3** (**Square-degree property $Sq$**). *Let $\{G_n\}_n$ denote a sequence of graphs. We say that $\{G_n\}_n$ exhibits the square-degree property if there exists some $c_1 > 0$ and $N_0 \in \mathbb{N}$ such that for all $n \geq N_0$ we have*

$$\sum deg \, v_{i,n}^2 \geq c_1 \left( \sum deg \, v_{i,n} \right)^2 \, .$$

*We denote the set of graph sequences satisfying the square-degree property by $S_q$, i.e. if $\{G_n\}_n$ satisfies $Sq$ then $\{G_n\}_n \in S_q$.*

We note that Cauchy-Schwarz inequality gives $c_1 = 1/n$, which is not satisfactory as we need a strictly positive lower bound $c_1 > 0$ for all $n$. The square-degree property says that the ratio between the sum of the degree squared and square of the sum of degrees is bounded from below as $n$ goes to infinity. As the degree of a node is either zero or positive, this cannot be satisfied if the degree distribution is uniform, because then the sum of the mixed product terms $deg \, v_{i,n} \times deg \, v_{j,n}$ would hold the bulk weight compared to the square

terms $(\deg v_{i,n})^2$, especially as there are $\binom{n}{2}$ mixed product terms and only $n$ square terms. Therefore, we expect a graph sequence satisfying this property to have some inequalities in the degree distribution. For example, it may contain a set of "big player" nodes with large degree values.

Using the square-degree property $Sq$ we characterize graph sequences $\{G_n\}_n$ as shown in Figure 3, in which the blue text represents results obtained in this paper. If a graph sequence converges in homomorphism density, then by Theorem 2.10 a graphon exists. In such instances, we consistently use $W$ and $U$ for graphons corresponding to $\{G_n\}_n$ and $\{H_m\}_m$ respectively. It is well-known that for converging dense graph sequences $\{G_n\}_n$, the graphon $W \neq 0$, while sparse graph sequences correspond to $W = 0$. This can be easily verified using the fact that for a converging graph sequence edge density and the non-zero area of the empirical graphon have the same limit.

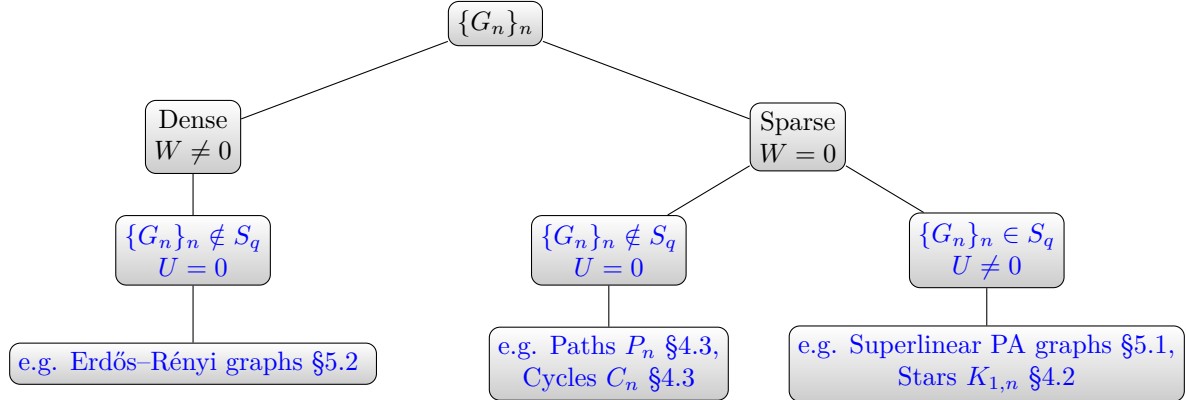

**Figure 3:** *Characterization of graph sequences $\{G_n\}_n$ with results discussed in this paper in blue text. $\{G_n\}_n \in S_q$ indicates that the graph sequences satisfies the square-degree property. If $\{G_n\}_n$ converges to $W$ (with respect to the homomorphism density), then for dense graphs $\{G_n\}_n$, $W \neq 0$, but $U = 0$. Recall that sparse graphs converge to $W = 0$. However if $\{G_n\}_n \in S_q$ and $\{H_m\}_m$ converges to $U$, then $U \neq 0$. For sparse $\{G_n\}_n \notin S_q$ then $U = 0$.*

We show that dense graph sequences do not satisfy the square-degree property $Sq$ in Section 3.2. If $\{G_n\}_n$ converges for dense sequences, then $\{H_m\}_m$ converges to $U = 0$, i.e., line graphs of dense graph sequences converge to the zero graphon. If $\{G_n\}_n$ is sparse then we know that $W = 0$ However, we cannot distinguish between different sparse graphs using $W$. We suppose $\{G_n\}_n$ converges to $W$ and find conditions under which $\{H_m\}_m$ converges to $U$ in Section 3.4. If the line graphs $\{H_m\}_m$ of sparse $\{G_n\}_n$ that satisfy $Sq$ converge, then $U$ can distinguish different types of sparse graphs. This means that line graphs of sparse graphs can be more revealing which we illustrate in Sections 4 and 5. The square-degree property $Sq$ is important because only graphs satisfying $Sq$ give rise to $U \neq 0$, if $\{H_m\}_m$ converges. Furthermore, not all sparse graphs satisfy $Sq$. Paths $P_n$ or cycles $C_n$ are such sparse graphs. Therefore, the subset of sparse graphs satisfying $Sq$ gives us certain types of graphs such as stars $K_{1,n}$ or superlinear preferential attachment graphs. For these graph sequences the line graphs converge to the limit $U \neq 0$. We will explore the square-degree property next.

### 3.2 Graph sequences with square-degree property $Sq$ are sparse

**Lemma 3.4.** *If $\{G_n\}_n \in S_q \implies \{G_n\}_n \in S$, i.e., graph sequences satisfying the square-degree property are sparse.*

*Proof.* As $\{G_n\}_n \in S_q$ there exist some $c_1 > 0$ and $N_0 \in \mathbb{N}$ such that for all $n \geq N_0$ we have

$$\sum \deg v_{i,n}^2 \geq c_1 \left(\sum \deg v_{i,n}\right)^2 .$$

As

$$n(n-1)^2 \geq \sum \deg v_{i,n}^2 \geq c_1 \left(\sum \deg v_{i,n}\right)^2 = 4c_1 m^2 , \tag{2}$$

we get $m \in O(n^{3/2})$ making $\{G_n\}_n$ sparse. From the above inequality we can see that

$$\limsup_{n \to \infty} \frac{m^2}{n^4} = \limsup_{n \to \infty} \frac{1}{4c_1 n} = 0 \,,$$

making $\lim_{n \to \infty} m/n^2 = 0$.  $\square$

Lemma 3.4 shows that the sparse graphs are a superset of graphs satisfying the square-degree property. However, not all sparse graphs satisfy $Sq$, for example paths and cycles. Therefore

$$S_q \subset S \,.$$

**Corollary 3.5.** *If $\{G_n\}_n \in D \implies \{G_n\}_n \notin S_q$, i.e., dense graph sequences do not satisfy the square-degree property.*

### 3.3 Only line graphs of graphs with square-degree property are dense

**Theorem 3.6.** *Let $\{G_n\}_n \in S$ be a sparse graph sequence. Let $\{H_m\}_m$ be the corresponding sequence of line graphs with $H_m = L(G_n)$. Then $\{G_n\}_n \in S_q \equiv \{H_m\}_m \in D$, i.e., $\{G_n\}_n$ satisfies $Sq$ if and only if $\{H_m\}_m$ is dense.*

*Proof.* 1. First we show $\{G_n\}_n \in S_q \implies \{H_m\}_m \in D$. Suppose $\{G_n\}_n \in S_q$ . Then from Definition 3.3 there exists some $c_1 > 0$ and $N_0 \in \mathbb{N}$ such that for all $n \geq N_0$ we have

$$\sum \deg v_{i,n}^2 \geq c_1 \left( \sum \deg v_{i,n} \right)^2 = 4c_1 m^2 \,,$$

where $m$ denotes the number of edges in $G_n$. From equation (1) the edge density of the line graph $L(G_n)$ is

$$\begin{aligned}
\text{density}(H_m) &= \frac{\frac{1}{2}\sum_i (\deg\ v_{i,n})^2 - m}{\frac{1}{2}m(m-1)} \,, \\
&\geq \frac{\frac{1}{2}4c_1 m^2 - m}{\frac{1}{2}m(m-1)} \,, \\
&= \frac{2c_1 - \frac{1}{m}}{\frac{1}{2} - \frac{1}{2m}} \,.
\end{aligned}$$

Thus,
$$\liminf_{m \to \infty} \text{density}(H_m) = 4c_1 > 0 \,.$$

2. Next we show $\{H_m\}_m \in D \implies \{G_n\}_n \in S_q$. If the line graphs $\{H_m\}_m$ are dense, i.e., $\{H_m\}_m \in D$ we have

$$\text{density}(H_m) = \frac{\frac{1}{2}\sum_i (\deg\ v_{i,n})^2 - m}{\frac{1}{2}m(m-1)} \geq c > 0 \quad \text{for all} \quad m > M_0 \in \mathbb{N}.$$

This can only happen when

$$\sum_i (\deg\ v_{i,n})^2 \geq c'm^2 \quad \text{where} \quad c' > 0 \,,$$

implying that $\{G_n\}_n$ satisfies the square-degree property.

$\square$

Next we explore graph sequences $\{G_n\}_n$ that do not satisfy $Sq$, i.e. $\{G_n\}_n \notin S_q$.

**Lemma 3.7.** *If $\{G_n\}_n$ does not satisfy the square-degree property, i.e., $\{G_n\}_n \notin S_q$, then*

$$\liminf_{m \to \infty} \operatorname{density}(H_m) = 0 \, .$$

*Additionally if the graph sequence $\{H_m\}_m$ is convergent in edge density, then*

$$\lim_{m \to \infty} \operatorname{density}(H_m) = 0 \, .$$

Lemma 3.7 coupled with Theorem 3.6 show that dense $\{H_m\}_m$ can only occur as a result of $\{G_n\}_n \in S_q$. This is shown in Figure 4 with the shaded area representing dense $\{H_m\}_m$.

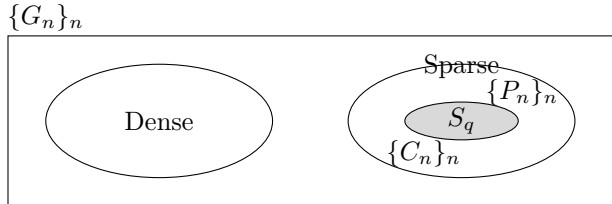

**Figure 4:** *The Euler diagram of the space of dense and sparse graph sequences, and indicate where there are graph sequences satisfying the square-degree property. The set $S \backslash S_q$ is non-empty as paths $\{P_n\}_n$, cycles $\{C_n\}_n$ and other graphs live here. The line graphs $\{H_m\}_m$ are dense in the shaded set $S_q$.*

### 3.4 Conditions for non-zero graphons of line graphs

In this section we explore graph sequences converging in homomorphism density. We suppose $\{G_n\}_n$ converges to $W$ and show that under the square-degree property, $\{H_m\}_m$ converges to a non-zero $U$. We will start with homomorphism densities.

#### 3.4.1 Revisiting graph homomorphisms

Recall when defining the empirical graphon (Definition 2.9) we divide the interval $[0,1]$ into $n$ equal subintervals $I_1, I_2, \ldots, I_n$ where each $I_j$ has length $1/n$. We use this construction in the next Lemma. Furthermore, recall that the homomorphism density $t(\bullet\!\!-\!\!\bullet, G_n) = 2m/n^2$ while the edge density, $\operatorname{density}(G_n) = 2m/(n(n-1))$ (Section 2.2.2) making the two densities converge to the same limit.

**Lemma 3.8.** *Let $H_m = L(G_n)$ and let $W_n$ be the empirical graphon of $G_n$ with $[0,1]$ divided into $n$ equal intervals $\{r_1, \ldots r_n\}$. Let $U_m$ be the empirical graphon of $H_m$ with $[0,1]$ equally divided into $m$ intervals $\{q_1, \ldots, q_m\}$. Then $t(\bullet\!\!-\!\!\bullet, H_m)$ can be written as*

$$t(\bullet\!\!-\!\!\bullet, H_m) = \sum_{i,j} U_m(q_i, q_j) \cdot \frac{1}{m^2} = \sum_{\substack{i,j,k \\ i \neq j}} W_n(r_i, r_k) W_n(r_k, r_j) \cdot \frac{1}{m^2} \, .$$

#### 3.4.2 Converging graph sequences

**Lemma 3.9.** *Let $\{G_n\}_n$ be a dense graph sequence converging to $W$ and let $H_m = L(G_n)$. Then $\{H_m\}_m$ converges to $U(x, y) = 0$ almost everywhere.*

Recall the definition of the cut-norm (Definition 2.5). The following lemma shows that for a graph sequence $\{G_n\}_n$ satisfying the square-degree property, if the sequence of line graphs $\{H_m\}_m$ converge to $U$, then $U$ has a strictly positive cut-norm. But Lemma 3.11 shows that for sparse graphs that do *not* have the square-degree property, the graphon corresponding to the line graph is uniformly zero.

**Lemma 3.10.** *Let $\{G_n\}_n \in S_q$ and let $H_m = L(G_n)$. If $\{H_m\}_m$ converges to $U$ then $U$ has strictly positive cut-norm, that is $\|U\|_\square > 0$.*

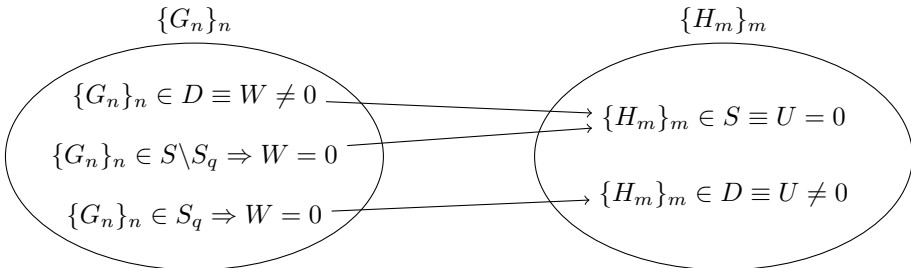

**Figure 5:** *The map from converging $\{G_n\}_n$ to converging $\{H_m\}_m$, summarising Lemmas 3.9, 3.10 and 3.11.*

**Lemma 3.11.** *Let $\{G_n\}_n \in S \backslash S_q$ and let $H_m = L(G_n)$. If $\{H_m\}_m$ converges to $U$, then $U = 0$ almost everywhere.*

For graph sequences $\{G_n\}_n$ converging in homomorphism density the Euler diagram of sparse and dense graphs is given in Figure 6.

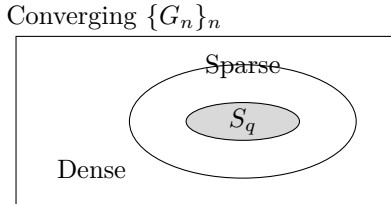

**Figure 6:** *The Euler diagram in Figure 4 updated for converging $\{G_n\}_n$.*

Lemmas 3.9, 3.10 and 3.11 can be used map different instances of $W$ to $U$ depending on the characteristics of $\{G_n\}_n$. For $W$ and $U$ to exist both sequences $\{G_n\}_n$ and $\{H_m\}_m$ need to converge. Figure 5 shows this relationship.

### 3.4.3 Orthogonal spaces

**Lemma 3.12.** *Suppose $\{G_n\}_n$ converges to $W$ and $\{H_m\}_m$ converges to $U$ where $H_m = L(G_n)$. Then the inner product*

$$\langle W, U \rangle = \int_{[0,1]^2} W(x,y)U(x,y)\,dxdy = 0\,.$$

*Thus, graphons $U$ obtained from line graphs are orthogonal to graphons $W$ with respect to the above inner product.*

## 4 Results for deterministic graphs

In this section, consider graph sequences consisting of disjoint star graphs. We show that although the original graph sequences $\{G_n\}_n$ are sparse, the corresponding sequences of line graphs $\{H_m\}_m$ converge to distinct non-zero graphons.

### 4.1 Dense line graphs, for star graphs

Consider a sequence of graphs $\{G_n\}_n$ as follows: For $n = 1$ we start with a single node $v_0$. At each step we add a node and connect it to $v_0$. At the $(n+1)$st step, this will give us a star graph $K_{1,n}$. Next we consider the line graph density of star graphs.

**Lemma 4.1.** *Let $\{G_n\}_n$ denote a sequence of star graphs i.e, $G_n = K_{1,n-1}$ and let $H_m = L(K_{1,n-1})$. Then $\{K_{1,n-1}\}_n \in S_q$. Moreover $\mathrm{density}(H_m) = 1$ and $\lim_{m\to\infty} \mathrm{density}(H_m) = 1$.*

*Proof.* Line graphs of star graphs are complete (Lemma 2.3-6). This gives us the desired result. An alternate proof from first principles is given in the Appendix. □

## 4.2  Graphons of line graphs of star graphs

Suppose $\{G_n\}_n$ is a sparse graph sequence. Note that $\{G_n\}_n$ converges to $W(x, y) = 0$ almost everywhere as per the cut-metric (Definition 2.6), $\|W_{G_n} - W\|_\square = \frac{2m}{n^2} \to 0$. As any sequence of sparse graphs converges to $W(x, y) = 0$, we cannot differentiate different types of sparse graphs from $W$. However, we can differentiate different types of sparse graphs using line graphs. In the following, we consider single and disjoint star graphs as an example of different sparse graphs.

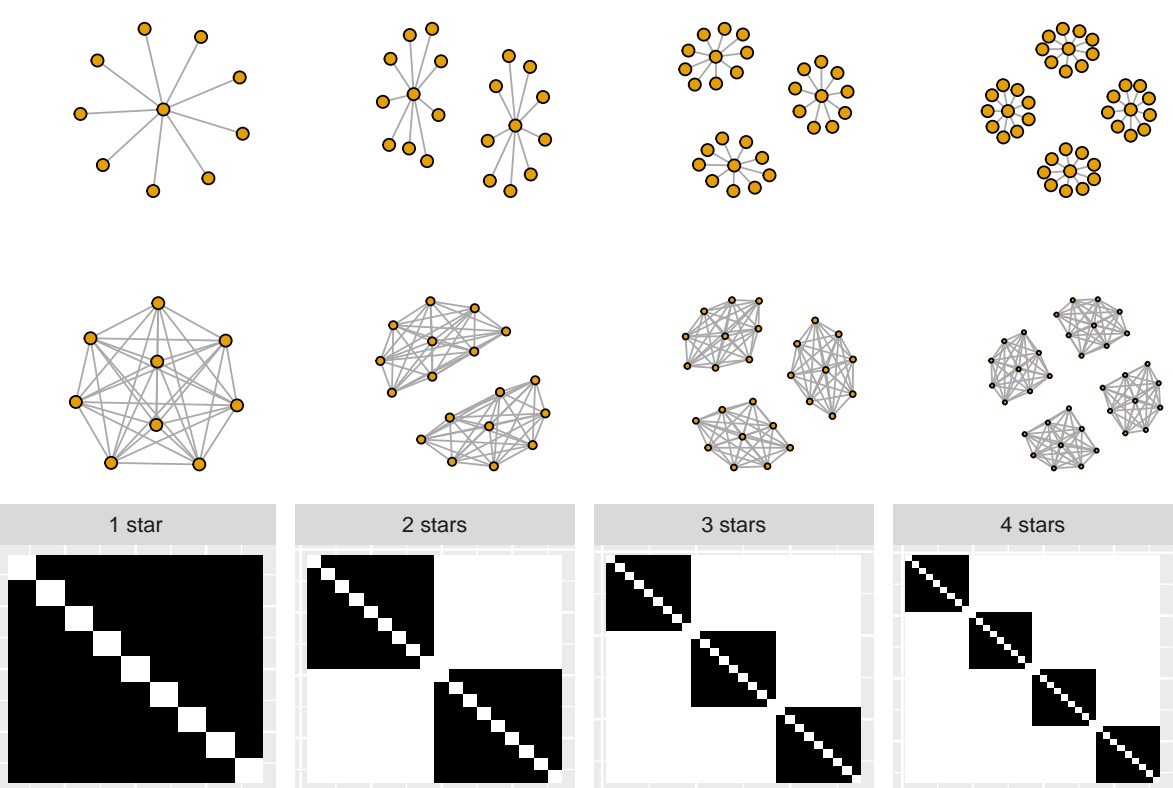

**Figure 7:** *Top row: Graphs of 1 to 4 disjoint stars. Recall the graphon $W = 0$ for star graphs. Middle row: Line graphs of disjoint stars in top row. Line graphs of star graphs are complete graphs. Bottom row: The empirical graphons of the line graphs $U_{H_m}$ of the star graphs shown on top.*

### 4.2.1  Single star graphs

Since the star graph $K_{1,n}$ is sparse, a sequence of star graphs converges to graphon $W = 0$. In the following lemma, we show that the corresponding sequence of line graphs $H_m = L(G_n)$ converge to a non-zero graphon $U$.

**Lemma 4.2.** *The line graphs $\{H_m\}_m$ of a sequence of star graphs $\{K_{1,n-1}\}_n$ satisfy*

$$\|U_{H_m} - U\|_\square = \frac{1}{m} \, ,$$

*where $U_{H_m}$ denotes the empirical graphon (Definition 2.9) of $H_m$ and $U(x, y) = 1$. Therefore, the line graphs of star graphs converge to the graphon $U$ in the cut metric (Definition 2.6).*

*Proof.* For $n \geq 2$ we consider the line graphs $H_m = L(K_{1,n-1})$ of this sequence. The line graph $H_m$ of a star graph $K_{1,n-1}$ is a complete graph $K_{n-1}$ (Lemma 2.3-6). We obtain the empirical graphon (Definition 2.9) of $H_m$ by splitting the interval $[0,1]$ into $m$ equal intervals $\{I_1, I_2, \ldots, I_m\}$ and for $x \in I_i, y \in I_j$ have

$$
U_{H_m}(x,y) = \begin{cases} 0 & \text{if } i = j, \\ 1 & \text{otherwise} \end{cases} .
$$

The empirical graphon $U_{H_m}$ is illustrated in the bottom leftmost diagram in Figure 7. Consider $U(x,y) = 1$ for all $x, y$. The cut norm (Definition 2.5) of $\|U_{H_m} - U\|_\square$ is

$$
\|U_{H_m} - U\|_\square = \sup_{S,T} \left| \int_{S \times T} U_{H_m}(x,y) - U(x,y) \, dx dy \right| .
$$

Using the intervals $\{I_1, I_2, \ldots, I_m\}$ and for $x \in I_i, y \in I_j$ we have

$$
U(x,y) - U_{H_m}(x,y) = \begin{cases} 1 & \text{if } i = j, \\ 0 & \text{otherwise} \end{cases} ,
$$

giving

$$
\|U_{H_m} - U\|_\square = \frac{1}{m^2} \times m = \frac{1}{m},
$$

as each $I_i \times I_i$ square would give rise to $\frac{1}{m^2}$ area. We have used $S = T = [0,1]$ as any $S \subset [0,1]$ and $T \subset [0,1]$ would give smaller area. The cut metric (Definition 2.6)

$$
\delta_\square(U_{H_m}, U) = \inf_\varphi \|U_{H_m} - U^\varphi\|_\square = \|U_{H_m} - U\|_\square,
$$

as $U^\varphi = U$ when $U(x,y) = 1$. As $\lim_{m \to \infty} \|U_{H_m} - U\|_\square = 0$, we have

$$
\lim_{m \to \infty} \delta_\square(U_{H_m}, U) = 0,
$$

and from Theorem 2.11 (Borgs et al., 2011) $\{H_m\}_m$ converges to $U$. We note that this works for any $U(x,y) = 1$ almost everywhere. □

### 4.2.2  Multiple stars

Next we consider $k$ disjoint stars denoted by $G_{n_i} = \{K_{1,s_1}, K_{1,s_2}, \ldots, K_{1,s_k}\}$ and the sequence $\{G_{n_i}\}_i$ as follows: When $i = 1$ we start with $k$ nodes each denoting the centre of a star. Let $\{r_1, \ldots, r_k\}$ denote positive integers and let $R = \sum_j r_j$. At each step we add $R$ nodes to the graph. Of the $R$ nodes, $r_j$ nodes connect to $K_{1,s_j}$ for $j \in \{1, \ldots k\}$. This process results in $k$ disjoint stars with the $j^{\text{th}}$ star having $1 + ir_j$ nodes at the $i^{\text{th}}$ step. The node ratios converge to $r_1 : r_2 : \ldots : r_k$ as $i$ goes to infinity. The following lemma shows that the line graphs of disjoint stars converge to an almost block diagonal graphon.

**Lemma 4.3.** *Let $\{G_{n_i}\}_i$ denote a disjoint set of $k$ star graphs $\{K_{1,s_1}, K_{1,s_2}, \ldots, K_{1,s_k}\}$ where $G_{n_i}$ has $n_i$ vertices and the number of degree-1 vertices of the stars satisfy the ratio $r_1 : r_2 : \ldots : r_k$ where each $r_j \in \mathbb{Z}^+$. Consider the graphon $U$ obtained by splitting the interval $[0,1]$ into $k$ sub intervals $\{I_1, I_2, \ldots, I_k\}$ such that the length of $I_r$ denoted by $L(I_r)$ satisfies the following: $L(I_1) : L(I_2) : \ldots : L(I_k) = r_1 : r_2 : \ldots : r_k$ and for $x \in I_i$ and $y \in I_j$*

$$
U(x,y) = \begin{cases} 1 & \text{if } i = j, \\ 0 & \text{otherwise} \end{cases} ,
$$

*making $U$ is a block diagonal graphon. The line graphs $H_{m_i} = L(G_{n_i})$ satisfy*

$$
\|U_{H_m} - U\|_\square = \frac{1}{m_i},
$$

*where $U_{H_m}$ denotes the empirical graphon (Definition 2.9) of $H_{m_i}$ making $\{H_{m_i}\}_i$ converge to the graphon $U$ in the cut metric (Definition 2.6).*

**Remark 4.4.** *Both single stars and multiple disjoint stars $\{G_n\}_n$ give rise to $W = 0$. However their line graphs $\{H_m\}_m$ give rise to different graphons $U$ as shown in Lemmas 4.2 and 4.3. This is an example of differentiating sparse graphs in the line graph space. See Figure 7.*

### 4.3 Line graphs of some dense and sparse graphs

Next, we go through some well known graphs and compute their line graph edge densities. We consider specific examples of graph sequences $\{G_n\}_n \in D$, and $\{G_n\}_n \in S \backslash S_q$.

**Theorem 4.5.** *Let $\{G_n\}_n$ be a sequence of graphs where $G_n$ has $n$ vertices and $m$ edges. Let $H_m = L(G_n)$ and suppose $m \to \infty$ as $n \to \infty$. Then $\{G_n\}_n$ with properties described below give rise to following line graph edge densities.*

1. *Suppose $G_n$ is the complete graph $K_n$. Then the edge density of the corresponding line graph, $\mathrm{density}(H_m) = \frac{4}{n+1}$ where $m = \frac{1}{2}n(n-1)$ and $\lim_{m \to \infty} \mathrm{density}(H_m) = 0$. Furthermore, $\{K_n\}_n \in D$ and $\{H_m\}_m \in S$.*

2. *Suppose $G_n$ is an $r$-regular graph. Then the edge density $\mathrm{density}(H_m) = \frac{2(r-1)}{m-1}$ and $\lim_{m \to \infty} density(H_m) = 0$. Furthermore $\{G_n\}_n, \{H_m\}_m \in S \backslash S_q$.*

3. *Suppose $G_n$ is a path. Then the edge density $\mathrm{density}(H_m) = \frac{2}{m}$ and $\lim_{m \to \infty} \mathrm{density}(H_m) = 0$. Furthermore $\{G_n\}_n, \{H_m\}_m \in S \backslash S_q$.*

4. *Suppose $G_n$ is a cycle. Then the edge density $\mathrm{density}(H_m) = \frac{2}{m-1}$ and $\lim_{m \to \infty} \mathrm{density}(H_m) = 0$. Furthermore $\{G_n\}_n, \{H_m\}_m \in S \backslash S_q$.*

### 4.4 Empirical Experiments on Estimating Graphons

In this section we compare graphs generated from different empirical graphons. Let $G_n$ denote a star $K_{1,n-1}$ with $n$ vertices and let $H_m = L(G_n)$. We consider the empirical graphons (see Definition 2.9) $W_{G_n}$ and $U_{H_m}$ where we consistently use $W$ and $U$ to denote graphons related to $G_n$ and $H_m$ respectively. We consider the set of $k$ disjoint stars as illustrated in Figure 7. We want to evaluate how well these empirical graphons can generate graphs with $kn$ vertices where $n = 100$ and $k \in \{2, 3, 4, 5\}$. That is, do graphs generated from $W_{G_n}$ resemble stars when $n$ increases? Similarly, do graphs generated from $U_{H_m}$ resemble line graphs of stars when $m$ increases?

To evaluate this, we generate (following Definition 2.7) $W$-random graphs from $W_{G_n}$ and $U$-random graphs $U_{H_m}$ with $kn$ vertices i.e., let $\hat{G}_W = \mathbb{G}(kn, W_{G_n})$ and $\hat{H}_U = \mathbb{G}(kn, U_{H_m})$. Noting we cannot compare $\hat{G}_W$ and $\hat{H}_U$ because $\hat{G}_W$ is in the space of original graphs whereas $\hat{H}_U$ is in the space of line-graphs, we consider the line graph of $\hat{G}_W$, that is, let $\hat{H}_W = L(\hat{G}_W)$. Then we have 3 graphs in the line graph space, the actual line graph $H = L(G_{kn})$, the estimated line graph of the $W$-random graph $\hat{H}_W$ and the estimated $U$-random graph $\hat{H}_U$. We compare different quantities derived from $H$, $\hat{H}_W$ and $\hat{H}_U$ for different $k$. These include the edge-density, the triangle-density, and the cosine similarity of the degree distributions of $\hat{H}_U$ and $\hat{H}_W$ with $H$.

Figure 8 shows the values obtained from $\hat{H}_U$, $\hat{H}_W$ and $H$ for a single star graph and Figure 9 shows the metrics for 2 stars. All 3 metrics are better for $\hat{H}_U$ compared to $\hat{H}_W$. Interestingly, the edge and triangle densities of $\hat{H}_U$ are slightly lower than those of $H$ in all instances. This is because $\hat{H}_U$ is sampled from $U_{H_m}$ which has empty squares along the diagonal, which are effectively closed or blacked out in the graphon $U$ (see empirical graphons in Figure 7). In these two scenarios we know that the shaded-area of $U_{H_m}$ is less than that of $U$, and as such slightly lower edge and triangle densities are expected.

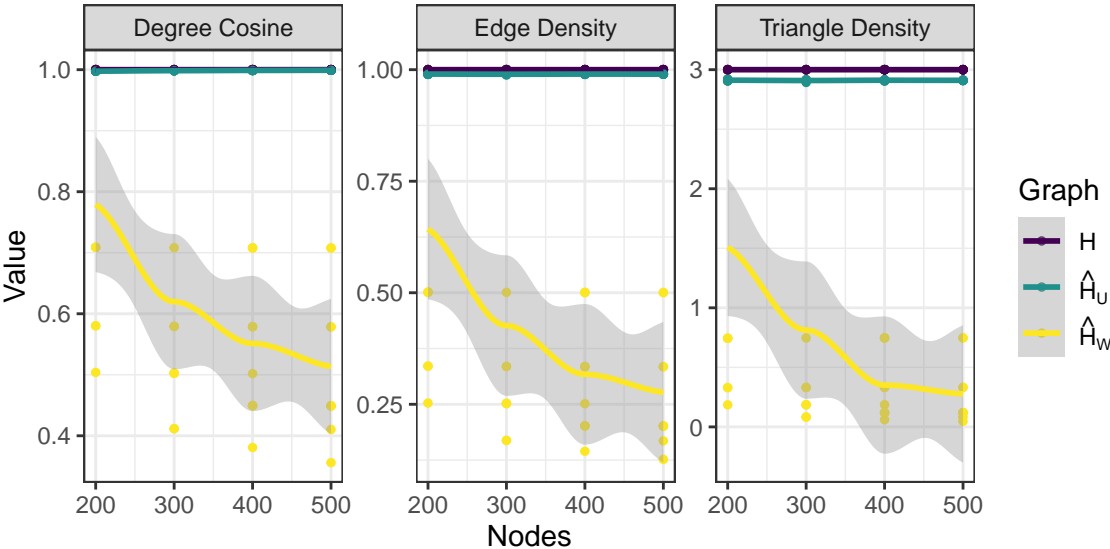

**Figure 8:** *Experiment with 1 star graph. Degree cosine similarity, edge density and triangle density for $H$, $\hat{H}_U$ and $\hat{H}_W$.*

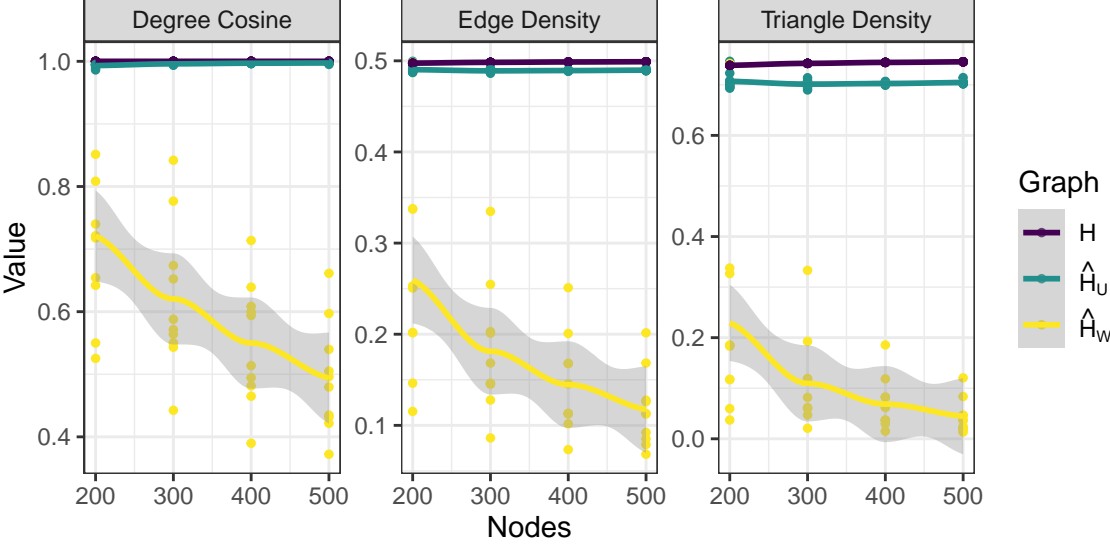

**Figure 9:** *Experiment with 2 star graphs. Degree cosine similarity, edge density and triangle density for $H$, $\hat{H}_U$ and $\hat{H}_W$.*

## 5 Results on probabilistic graphs

In this section, we consider two models of graphs that are popular in social network analysis, namely preferential attachment models (Albert & Barabási, 2002) and Erdős–Rényi model $G(n,p)$. As noted in Figure 3, the two models have contrasting behaviours. We show in Section 5.1 that superlinear preferential attachment models satisfy the square-degree property almost surely, and the corresponding line graphs $H_m$ converges to a non-zero graphon $U$. The Erdős–Rényi model results in a dense non-zero graphon $W$, and we show in Section 5.2 that the edge density of $H_m$ convergence to zero with an exponential rate.

### 5.1 Superlinear preferential attachment graphs

Preferential attachment models (Albert & Barabási, 2002) consider nodes connecting to more connected nodes with higher probability. Specifically the probability $\Pi(i)$ that a new node connects to node $i$, which has degree $k_i$ is given by

$$\Pi(i) = \frac{k_i^\alpha}{\sum_i k_i^\alpha}\,, \tag{3}$$

where $\alpha$ is a parameter. The three regimes $\alpha < 1$, $\alpha = 1$ and $\alpha > 1$ are called sublinear, linear and superlinear preferential attachment respectively. Suppose we start with $s_0$ nodes and $s_0$ edges and at each time step $t$ a new node is added to the network with $s$ edges. After $t$ timesteps the network has

$$n = t + s_0 \quad \text{nodes and} \quad m = s_0 + ts \quad \text{edges.} \tag{4}$$

For growing networks with superlinear preferential attachment Krapivsky & Redner (2001); Krapivsky et al. (2000) state that the maximum degree $k_{\max}$ satisfies

$$k_{\max} \sim n\,.$$

Sethuraman & Venkataramani (2019) prove a more rigorous version of the above statement. They show that,

$$P\left[\lim_{n\to\infty} \frac{1}{n}k_{\max} = 1\right] = 1\,.$$

We will use this result to show that superlinear preferential attachment graphs satisfy the square-degree property almost surely.

**Lemma 5.1.** *Let $\{G_n\}_n$ denote a sequence of graphs growing by superlinear preferential attachment satisfying equation (3) with $\alpha > 1$. Then $\{G_n\}_n \in S_q$ almost surely.*

*Proof.* Using the result from Sethuraman & Venkataramani (2019) we know that for every $\epsilon > 0$ there exists $N_0 \in \mathbb{N}$ such that

$$P\left[\left|\frac{1}{n}k_{\max} - 1\right| < \epsilon\right] = 1\,,$$

for all $n > N_0$. That is, almost surely

$$1 - \epsilon < \frac{1}{n}k_{\max} < 1 + \epsilon\,,$$

for $n > N_0$. Rearranging the equations for $n$ and $m$ (equation (4)) we get $ns = m + (s-1)s_0$ giving us $ns > m$. Hence,

$$\sum (\deg v_i^2) > k_{\max}^2 > (1-\epsilon)^2 n^2 > \frac{(1-\epsilon)^2 m^2}{s^2} \quad \text{almost surely.}$$

Thus, for $n > N_0$

$$P\left[\sum(\deg v_i^2) > \frac{(1-\epsilon)^2}{s^2}m^2\right] = 1$$

showing that superlinear preferential attachment graphs satisfy the square-degree property (Definition 3.3) almost surely for large values of $n$. From Theorem 3.6 they produce dense line graphs. If $\{t(F, H_m)\}_m$ converges for all graphs $F$, where $H_m = L(G_n)$ then Theorem 2.10 (Borgs et al., 2008) ensures $\{H_m\}_m$ converges to a graphon $U$. As $\{H_m\}_m$ is dense $U \neq 0$. □

### 5.2 Erdős–Rényi graphs

The Erdős–Rényi model $G(n, p)$ describes graphs of $n$ vertices with edge probability $p$, where $p$ is a parameter. Each edge is equally likely to be included in the graph. The degree distribution for any vertex in $G_n \sim G(n, p)$ is binomial with parameters $n - 1$ and $p$. For a given $n$ and $p$ there exists a graph distribution as different edges can be included or left out in different graphs. Expectations are calculated with respect to this graph distribution.

**Theorem 5.2.** *Let $G_n$ be an Erdős–Rényi graph sampled from a $G(n, p)$ model and suppose $G_n$ has $n$ nodes and $m$ edges. Let $H_m = L(G_n)$. Then for any $c \in (0, 1)$, the edge density of $H_m$ satisfies*

$$P\left[density(H_m) \geq c\right] \leq \exp\left(-\frac{\alpha^2 pn(n-1)}{4}\right) + \exp\left(\ln n - \frac{\beta^2 p(n-1)}{3}\right) + \exp\left(-\frac{\alpha^2 pn(n-1)}{6}\right),$$

*where $\alpha \in (0, 1)$, $n > \frac{4}{c(1-\alpha)^2}$ and $\beta = \frac{\sqrt{cn}(1-\alpha)}{2} - 1$. Therefore, as $n$ and $m$ go to infinity the edge density of $H_m$ satisfies*

$$\lim_{m \to \infty} P\left[density(H_m) = 0\right] = 1.$$

## 6  Conclusions

Graphons are a compact representation or a graph model that can generate arbitrarily large graphs. The standard construction of the graphon is useful for dense graphs, but sparse graphs converge to the zero graphon, limiting its utility. The classical construction concerns the non-zero area of the graphon, which is zero for sparse graphs. To overcome this limitation, methods have been proposed that can capture and differentiate point masses, a feature of sparse graphs. Typically, these methods have strong measure-theoretic underpinnings and often involve complex mathematical machinery. In this paper, we show that for a subset of sparse graphs, taking the line graph gives promising results. We propose a condition on sparse graphs, called the square-degree property, which results in dense line graphs. This enables standard graph convergence to be used to analyse graph limits.

We show that graphs that satisfy the square-degree property are sparse, but map to dense line graphs, while graphs that do not satisfy the square-degree property give rise to sparse line graphs. Using the square degree property, we illustrate three cases. First we show that star graphs are sparse and converge to the zero graphon ($W = 0$). However, line graphs of star graphs are complete and converge to the graphon $U = 1$. Similarly, multiple star graphs converge to $W = 0$, but their line graphs converge to a block diagonal graphon $U \neq 0$. Thus, line graphs of multiple star graphs (since they satisfy the the square-degree property) are dense, making the graphon of these line graphs non-zero when convergence exists. Second we show that preferential attachment models give rise to graph sequences that satisfy the square degree property, and hence result in line graphs that converge to non-zero graphons. Third we prove that Erdős–Rényi graphs almost surely give rise to sparse line graphs. We hope that this new approach of using line graphs to analyse graph limits provides an interesting tool for researchers working on graphons.

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

## A    Proofs on sparse graphs with dense line graphs

**Corollary A.1.** *If $\{G_n\}_n \in D \implies \{G_n\}_n \notin S_q$, i.e., dense graph sequences do not satisfy the square-degree property.*

*Proof.* As $D \subset \bar{S}$, where $\bar{S}$ denotes the complement of $S$, this is true because of the contrapositive of Lemma 3.4. It can be quickly verified that dense graph sequences do not satisfy equation (2) in Lemma 3.4 because for dense graphs $m \geq cn^2$ for some $c > 0$. $\qquad\square$

**Lemma 3.7.** *If $\{G_n\}_n$ does not satisfy the square-degree property, i.e., $\{G_n\}_n \notin S_q$, then*

$$\liminf_{m\to\infty} \text{density}(H_m) = 0\,.$$

*Additionally if the graph sequence $\{H_m\}_m$ is convergent in edge density, then*

$$\lim_{m\to\infty} \text{density}(H_m) = 0\,.$$

*Proof.* The first part is the contra-positive of Theorem 3.6(2). We prove it from first principles for the sake of completeness. Let us restate the square-degree property and consider its negation. If a graph sequence $\{G_n\}_n$ satisfies the square-degree property, then there exists constants $c_1 \in (0,1)$ and $N_0 \in \mathbb{N}$ such that for all $n \geq N_0$ we have

$$\sum \deg v_{i,n}^2 \geq c_1 \left(\sum \deg v_{i,n}\right)^2\,.$$

The negation of square-degree property, $\neg Sq$ says that for all $c_1 > 0$ and $N_0 \in \mathbb{N}$ there exists $n \geq N_0$ such that

$$\sum \deg v_{i,n}^2 < c_1 \left(\sum \deg v_{i,n}\right)^2\,.$$

For every $N_0 \in \mathbb{N}$ there exists $n \geq N_0$ such that this inequality is satisfied. Consider

$$A_{c_1} = \left\{n \in \mathbb{N} : n \geq N_0, \sum \deg v_{i,n}^2 < c_1 \left(\sum \deg v_{i,n}\right)^2\right\}\,.$$

If $|A_{c_1}|$ was finite, then we can pick $N_\nu = \max(A_{c_1}) + 1$ and for $n \geq N_\nu$ the inequality $\sum \deg v_{i,n}^2 < c_1 \left(\sum \deg v_{i,n}\right)^2$ would not be satisfied. Thus, the set $A_{c_1}$ has infinitely many elements. Therefore for every $c_1 \in (0,1)$ and $N_0 \in \mathbb{N}$ there is an infinite sequence $A_{c_1}$ such that for any $n \in A_{c_1}$

$$\sum \deg v_{i,n}^2 < c_1 \left(\sum \deg v_{i,n}\right)^2\,.$$

Hence we can consider a sequence of sequences $\{A_{c_{1_i}}\}_{c_{1_i}}$ where $c_{1_i} > c_{1_j}$ when $i < j$. From this sequence set we can choose a diagonal subsequence $\{n_1, n_2, \ldots\}$ such that $n_1 \in A_{c_{1_1}}$ and $n_2 \in A_{c_{1_2}}$ and so on, such that this sequence converges to zero. From equation (1) recall that

$$\text{density}(H_m) = \frac{\frac{1}{2}\sum_i (\deg\, v_{i,n})^2 - m}{\frac{1}{2}m(m-1)}\,.$$

For the diagonal subsequence selected above

$$\frac{\sum \deg v_{i,n}^2}{\left(\sum \deg v_{i,n}\right)^2} = \frac{\sum \deg v_{i,n}^2}{4m^2} \to 0\,,$$

giving us

$$\liminf_{m\to\infty} \text{density}(H_m) = 0 \,.$$

If $\{H_m\}_m$ is convergent, then all subsequences converge to the same limit and we get

$$\lim_{m\to\infty} \text{density}(H_m) = 0 \,.$$

$\square$

**Lemma 3.8.** *Let $H_m = L(G_n)$ and let $W_n$ be the empirical graphon of $G_n$ with $[0,1]$ divided into $n$ equal intervals $\{r_1, \ldots r_n\}$. Let $U_m$ be the empirical graphon of $H_m$ with $[0,1]$ equally divided into $m$ intervals $\{q_1, \ldots, q_m\}$. Then $t(\bullet\!\!-\!\!\bullet, H_m)$ can be written as*

$$t(\bullet\!\!-\!\!\bullet, H_m) = \sum_{i,j} U_m(q_i, q_j) \cdot \frac{1}{m^2} = \sum_{\substack{i,j,k \\ i \neq j}} W_n(r_i, r_k) W_n(r_k, r_j) \cdot \frac{1}{m^2} \,.$$

*Proof.* From Definition 2.4

$$t(F, H_m) = \sum_{\phi: V(F) \to V(H_m)} \prod_{ij \in E(F)} \beta_{\phi(i)\phi(j)}(H_m) \cdot \frac{1}{m^{|V(F)|}} \,,$$

where $\phi$ is a mapping from $V(F)$ to $V(H_m)$ and $\beta_{ij}(H_m)$ denotes the weight of edge $ij$ in graph $H_m$, which is either 1 or 0. Thus,

$$t(\bullet\!\!-\!\!\bullet, H_m) = \sum_{\phi: V(\bullet\!\!-\!\!\bullet) \to V(H_m)} \prod_{ij \in E(\bullet\!\!-\!\!\bullet)} \beta_{\phi(i)\phi(j)}(H_m) \cdot \frac{1}{m^2} \,,$$

$$= \sum_{\substack{\phi: V(\bullet\!\!-\!\!\bullet) \to V(H_m) \\ ij \in E(\bullet\!\!-\!\!\bullet)}} \beta_{\phi(i)\phi(j)}(H_m) \cdot \frac{1}{m^2} \,,$$

where we have dropped the product term as there is only one edge. We can replace the edge weight $\beta_{\phi(i)\phi(j)}(H_m)$ with the associated value in the empirical graphon $U_m(q_{\phi(i)}, q_{\phi(j)})$ giving us

$$t(\bullet\!\!-\!\!\bullet, H_m) = \sum_{\substack{\phi: V(\bullet\!\!-\!\!\bullet) \to V(H_m) \\ ij \in E(\bullet\!\!-\!\!\bullet)}} U_m(q_{\phi(i)}, q_{\phi(j)}) \cdot \frac{1}{m^2} \,,$$

$$= \sum_{i,j} U_m(q_i, q_j) \cdot \frac{1}{m^2} \,,$$

as $\phi$ can map the edge to any two vertices in $H_m$. Every edge in $G_n$ is mapped to a vertex in $H_m$ and 2 vertices in $H_m$ are connected if the corresponding edges in $G_n$ have a common vertex. That is, $L(\bullet\!\!-\!\!\bullet\!\!-\!\!\bullet) = \bullet\!\!-\!\!\bullet$, and for every edge in $H_m$ there is a corresponding set of two edges with a common vertex ($\bullet\!\!-\!\!\bullet\!\!-\!\!\bullet$) in $G_n$. As a result the empirical graphon (Definition 2.9),

$$U_m(q_i, q_j) = 1 \quad \text{if and only if} \quad W_n(r_k, r_\ell) W_n(r_\ell, r_s) = 1$$

for some $k, \ell, s \in \{1, \ldots, n\}$ with $k \neq s$. The reason $k \neq s$ is because we need 2 distinct edges in $G_n$ with a common vertex to make an edge in $H_m$. As a result of this one-to-one and onto mapping we have

$$\sum_{i,j} U_m(q_i, q_j) = \sum_{\substack{k,\ell,s \\ k \neq s}} W_n(r_k, r_\ell) W_n(r_\ell, r_s)$$

giving us the desired result. $\square$

**Lemma 3.9.** *Let $\{G_n\}_n$ be a dense graph sequence converging to $W$ and let $H_m = L(G_n)$. Then $\{H_m\}_m$ converges to $U(x,y) = 0$ almost everywhere.*

*Proof.* As $\{G_n\}_n$ is a dense graph sequence converging to $W$

$$\lim_{n\to\infty} t(\bullet\!\!-\!\!\bullet, G_n) = \lim_{n\to\infty} \frac{2m}{n^2} = c > 0 . \tag{5}$$

We will use this limit later. Let $W_n$ be the empirical graphon of $G_n$ with $[0,1]$ divided into $n$ equal intervals $\{r_1, \ldots r_n\}$ and let $U_m$ be the empirical graphon of $H_m$ with $[0,1]$ equally divided into $m$ intervals $\{q_1, \ldots, q_m\}$. The homomorphism density $\{t(\bullet\!\!-\!\!\bullet\!\!-\!\!\bullet, G_n)\}_n$ is a converging sequence as $\{G_n\}_n$ converges to $W$. We have

$$t(\bullet\!\!-\!\!\bullet\!\!-\!\!\bullet, G_n) = \sum_{i,j,k} W(r_i, r_k) W(r_k, r_j) \cdot \frac{1}{n^3} ,$$

converging as $n$ goes to infinity. From Lemma 3.8 we know

$$t(\bullet\!\!-\!\!\bullet, H_m) = \sum_{i,j} U(q_i, q_j) \cdot \frac{1}{m^2} ,$$

$$= \sum_{\substack{i,j,k \\ i \neq j}} W(r_i, r_k) W(r_k, r_j) \cdot \frac{1}{m^2} ,$$

$$\leq \sum_{i,j,k} W(r_i, r_k) W(r_k, r_j) \cdot \frac{1}{m^2} ,$$

$$= \sum_{i,j,k} \frac{1}{n^4} W(r_i, r_k) W(r_k, r_j) \cdot \frac{1}{\left(\frac{m}{n^2}\right)^2} .$$

As $n$ and $m$ go to infinity we get

$$\limsup_{m\to\infty} t(\bullet\!\!-\!\!\bullet, H_m) = \lim_{\substack{n\to\infty \\ m\to\infty}} \frac{1}{n} \cdot t(\bullet\!\!-\!\!\bullet\!\!-\!\!\bullet, G_n) \cdot \frac{1}{\left(\frac{m}{n^2}\right)^2} = 0 ,$$

as $m/n^2$ goes to $c/2 > 0$ (equation (5)) and $t(\bullet\!\!-\!\!\bullet\!\!-\!\!\bullet, G_n)$ converges. As $t(\bullet\!\!-\!\!\bullet, H_m)$ lies between 0 and 1 we get

$$\lim_{m\to\infty} t(\bullet\!\!-\!\!\bullet, H_m) = 0 .$$

As $t(\bullet\!\!-\!\!\bullet, H_m) = \frac{2m'}{n'^2}$ goes to 0, where $m'$ and $n'$ denote the number of edges and vertices in $H_m$, the cut-norm (Definition 2.5) satisfies

$$\|U_{H_m} - U\|_\square = \|U_{H_m}\|_\square = \frac{2m'}{n'(n'-1)} \to 0 ,$$

where $U(x,y) = 0$. As the cut-metric (Definition 2.6)

$$\delta_\square \left( U_{H_m}, U \right) = \inf_\varphi \|U_{H_m} - U^\varphi\| ,$$

$\{H_m\}_m$ converges to $U$ in the cut-metric as the infimum is considered and as $U^\varphi = U$ for $U = 0$. $\qquad\square$

**Lemma 3.10.** *Let $\{G_n\}_n \in S_q$ and let $H_m = L(G_n)$. If $\{H_m\}_m$ converges to $U$ then $U$ has strictly positive cut-norm, that is $\|U\|_\square > 0$.*

*Proof.* From Theorem 3.6 we know $\{G_n\}_n \in S_q \equiv \{H_m\}_m \in D$. Additionally, if $\{H_m\}_m$ converges to $U$ then $\{t(\bullet\!\!-\!\!\bullet, H_m)\}_m$ converges to $t(\bullet\!\!-\!\!\bullet, U)$. As $t(\bullet\!\!-\!\!\bullet, H_m) = \frac{2m'}{n'^2}$ where $m'$ and $n'$ denote the number of edges and nodes in $H_m$ where $n' = m$, the sequence $\frac{2m'}{n'^2}$ converges to some constant $c$. But as $\{H_m\}_m \in D$

$$t(\bullet\!\!-\!\!\bullet, U) = \lim_{n'\to\infty} \frac{2m'}{n'^2} = c > 0 ,$$

that is, the edge density of $\{H_m\}_m$ converges to a positive constant. The homomorphism density $t(\bullet\!\bullet, U)$ (Definition 2.4) is given by

$$t(\bullet\!\bullet, U) = \int_{[0,1]^2} U(x,y)\,dxdy\,,$$

which is equal to the cut-norm of $U$

$$\|U\|_\square = \sup_{S,T} \left| \int_{S \times T} U(x,y)dxdy \right|\,,$$

because $U(x,y) \in [0,1]$ and the supremum is achieved when $S = T = [0,1]$, giving us

$$\|U\|_\square = t(\bullet\!\bullet, U) > 0\,.$$

$\square$

**Lemma 3.11.** *Let $\{G_n\}_n \in S \backslash S_q$ and let $H_m = L(G_n)$. If $\{H_m\}_m$ converges to $U$, then $U = 0$ almost everywhere.*

*Proof.* As $\{G_n\}_n \in S$, it is sparse and it converges to $W = 0$. From Lemma 3.7 if $\{G_n\}_n \notin S_q$

$$\liminf_{m \to \infty} \text{density}(H_m) = 0\,.$$

As $\{H_m\}_m$ converges to $U$, the edge densities converge and we get $\lim_{m\to\infty} \text{density}(H_m) = 0$. The empirical graphon $U_{H_m}$ converges to $U$ and we have the cut norm (Definition 2.5) of the empirical graphon

$$\|U_{H_m}\| = \frac{2m'}{n'(n'-1)} \to 0$$

giving us

$$\lim_{m\to\infty} \|U_{H_m} - U\| = 0\,,$$

where $U = 0$. As the cut-metric (Definition 2.6)

$$\delta_\square(U_{H_m}, U) = \inf_\varphi \|U_{H_m} - U^\varphi\|\,,$$

we get the result. $\square$

**Lemma 3.12.** *Suppose $\{G_n\}_n$ converges to $W$ and $\{H_m\}_m$ converges to $U$ where $H_m = L(G_n)$. Then the inner product*

$$\langle W, U \rangle = \int_{[0,1]^2} W(x,y)U(x,y)\,dxdy = 0\,.$$

*Thus, graphons $U$ obtained from line graphs are orthogonal to graphons $W$ with respect to the above inner product.*

*Proof.* For converging sequences $\{G_n\}_n$ and $\{H_m\}_m$ we have $W = 0$ or $U = 0$ (Lemmas 3.9, 3.10 and 3.11). The graphon $W \neq 0$ only when $\{G_n\}_n \in D$. When $\{G_n\}_n \in D$ we have $\{H_m\}_m \in S$ giving $U = 0$. The graphon $U \neq 0$ only when $\{G_n\}_n \in S_q$ implying $W = 0$ as $S_q \subset S$. $\square$

# B  Proofs on results for deterministic graphs

**Lemma 4.1.** *Let $\{G_n\}_n$ denote a sequence of star graphs i.e, $G_n = K_{1,n-1}$ and let $H_m = L(K_{1,n-1})$. Then $\{K_{1,n-1}\}_n \in S_q$. Moreover $\text{density}(H_m) = 1$ and $\lim_{m\to\infty} \text{density}(H_m) = 1$.*

*Proof.* We present an alternate proof from first principles. For the sake of completeness, we do the computation from first principles. For a star graph

$$\deg v_i = \begin{cases} n-1 & \text{for star vertex }, \\ 1 & \text{otherwise} \end{cases}$$

giving us

$$\sum \deg v_{i,n}^2 = (n-1)^2 + 1 + \cdots + 1\,,$$
$$= (n-1)^2 + (n-1)\,,$$
$$\sum \deg v_{i,n} = m = n-1\,,$$
$$\frac{\sum \deg v_{i,n}^2}{\left(\sum \deg v_{i,n}\right)^2} = 1 + \frac{1}{n-1} > 1\,,$$

showing that $\{K_{1,n}\}_n \in S_q$ (Definition 3.3). From equation (1), the density of $H_m$ is given by

$$\text{density}(H_m) = \frac{\frac{1}{2}\sum_i (\deg v_{i,n})^2 - m}{\frac{1}{2}m(m-1)}\,,$$
$$= \frac{\frac{1}{2}((n-1)^2 + 1 + 1 + \ldots + 1) - (n-1)}{\frac{1}{2}(n-1)(n-2)}\,,$$
$$= \frac{\frac{1}{2}((n-1)^2 + (n-1)) - (n-1)}{\frac{1}{2}(n-1)(n-2)}\,,$$
$$= \frac{\frac{1}{2}(n-1)(n) - (n-1)}{\frac{1}{2}(n-1)(n-2)}\,,$$
$$= \frac{\frac{1}{2}(n-1)(n-2)}{\frac{1}{2}(n-1)(n-2)}\,,$$
$$= 1\,.$$

Thus, $\lim_{m\to\infty} \text{density}(H_m) = 1$. □

**Lemma 4.3.** *Let $\{G_{n_i}\}_i$ denote a disjoint set of $k$ star graphs $\{K_{1,s_1}, K_{1,s_2}, \ldots, K_{1,s_k}\}$ where $G_{n_i}$ has $n_i$ vertices and the number of degree-1 vertices of the stars satisfy the ratio $r_1 : r_2 : \ldots : r_k$ where each $r_j \in \mathbb{Z}^+$. Consider the graphon $U$ obtained by splitting the interval $[0,1]$ into $k$ sub intervals $\{I_1, I_2, \ldots, I_k\}$ such that the length of $I_r$ denoted by $L(I_r)$ satisfies the following: $L(I_1) : L(I_2) : \ldots : L(I_k) = r_1 : r_2 : \ldots : r_k$ and for $x \in I_i$ and $y \in I_j$*

$$U(x,y) = \begin{cases} 1 & \text{if } i = j \\ 0 & \text{otherwise} \end{cases}\,,$$

*making $U$ is a block diagonal graphon. The line graphs $H_{m_i} = L(G_{n_i})$ satisfy*

$$\|U_{H_m} - U\|_{\square} = \frac{1}{m_i}\,,$$

*where $U_{H_m}$ denotes the empirical graphon (Definition 2.9) of $H_{m_i}$ making $\{H_{m_i}\}_i$ converge to the graphon $U$ in the cut metric (Definition 2.6).*

*Proof.* The line graph of $k$ disjoint stars is $k$ disjoint complete subgraphs. This follows from Lemma 2.3 (4 and 6) as vertices of 2 different stars are not connected. Noting $H_{m_i}$ has $m_i$ vertices, we obtain the empirical graphon (Definition 2.9) of $H_{m_i}$ by splitting the interval $[0,1]$ into $m_i$ equal intervals $\{I_1, I_2, \ldots, I_{m_i}\}$.

At the $i$th step, the $j$th star $K_{1,s_j}$ has $1 + ir_j$ nodes and $ir_j$ edges. Then the corresponding complete subgraph $K_{s_j}$ of the line graph $H_{m_i}$ has $ir_j$ nodes as each node in the line graph corresponds to an edge in $G_{n_i}$. We

label nodes belonging to a complete subgraph consecutively. That gives us vertices $1, \ldots, ir_1$ corresponding to the first complete subgraph $K_{s_1}$, and nodes $(ir_1 + 1), \ldots, (ir_1 + ir_2)$ corresponding the second complete subgraph $K_{s_2}$ and so on. The ratio between the number of nodes in each subgraph is $r_1 : r_2 : \ldots : r_k$.

Let us group the vertices in $H_m$, $\{1, 2, \ldots, m_i\}$ into $k$ groups $\{J_1, J_2, \ldots, J_k\}$ according to the complete subgraph they belong to. Then for $x \in I_j, y \in I_h$ we have the empirical graphon (Definition 2.9) of $H_m$

$$
U_{H_{m_i}}(x, y) = \begin{cases} 1 & \text{if} \quad j, h \in J_\ell \text{ for some } \ell \text{ but } j \neq h \\ 0 & \text{if} \quad j = h \text{ as there are no loops} \\ 0 & \text{if} \quad j \in J_p \text{ and } h \in J_q \text{ where } p \neq q \end{cases} .
$$

The bottom row in Figure 7 shows empirical graphons for $k \in \{2, 3, 4\}$. Note that $U$ is a block diagonal graphon similar to $U_{H_{m_i}}$ differing to $U_{H_{m_i}}$ only on the diagonal. One can visualize $U$ by colouring the white squares on the diagonal in empirical graphons in Figure 7 for $k \in \{2, 3, 4\}$.

Then, the cut-norm (Definition 2.5),

$$
\|U_{H_m} - U\|_\square = \sup_{S,T} \left| \int_{S \times T} U_{H_m}(x, y) - U(x, y) \, dxdy \right| ,
$$
$$
= \frac{1}{m_i^2} \times m_i = \frac{1}{m_i} ,
$$

where we have used $S = T = [0, 1]$ in computing the cut-norm as any other $S$ or $T$ would give smaller area. Then the cut metric (Definition 2.6)

$$
\delta_\square(U_{H_m}, U) = \inf_\varphi \|U_{H_m} - U^\varphi\|_\square \leq \|U_{H_m} - U\|_\square = \frac{1}{m_i} .
$$

As $\|U_{H_m} - U\|_\square$ goes to zero as $m$ goes to infinity $\delta_\square(U_{H_m}, U)$ converges to zero. From Theorem 2.11 (Borgs et al., 2011) $\{H_{m_i}\}_{m_i}$ converges to $U$. □

**Theorem 4.5.** *Let $\{G_n\}_n$ be a sequence of graphs where $G_n$ has n vertices and m edges. Let $H_m = L(G_n)$ and suppose $m \to \infty$ as $n \to \infty$. Then $\{G_n\}_n$ with properties described below give rise to following line graph edge densities.*

1. *Suppose $G_n$ is the complete graph $K_n$. Then the edge density of the corresponding line graph, $\text{density}(H_m) = \frac{4}{n+1}$ where $m = \frac{1}{2}n(n-1)$ and $\lim_{m \to \infty} \text{density}(H_m) = 0$. Furthermore, $\{K_n\}_n \in D$ and $\{H_m\}_m \in S$.*

2. *Suppose $G_n$ is an r-regular graph. Then the edge density $\text{density}(H_m) = \frac{2(r-1)}{m-1}$ and $\lim_{m \to \infty} density(H_m) = 0$. Furthermore $\{G_n\}_n, \{H_m\}_m \in S \backslash S_q$.*

3. *Suppose $G_n$ is a path. Then the edge density $\text{density}(H_m) = \frac{2}{m}$ and $\lim_{m \to \infty} \text{density}(H_m) = 0$. Furthermore $\{G_n\}_n, \{H_m\}_m \in S \backslash S_q$.*

4. *Suppose $G_n$ is a cycle. Then the edge density $\text{density}(H_m) = \frac{2}{m-1}$ and $\lim_{m \to \infty} \text{density}(H_m) = 0$. Furthermore $\{G_n\}_n, \{H_m\}_m \in S \backslash S_q$.*

*Proof.* Recall that

$$
\text{density}(H_m) = \frac{\frac{1}{2} \sum (\deg\ v^2) - m}{\frac{1}{2}m(m-1)} .
$$

1. Suppose $G_n$ is the complete graph $K_n$. As density$(K_n) = 1$, the sequence $\{K_n\}_n$ is dense, i.e, $\{K_n\}_n \in D$. For $K_n$, deg $v_i = n - 1$ and $m = n(n-1)/2$ giving us

$$
\begin{aligned}
\text{density}(H_m) &= \frac{\frac{1}{2}n(n-1)^2 - \frac{1}{2}n(n-1)}{\frac{1}{2}\frac{1}{2}n(n-1)(\frac{1}{2}n(n-1) - 1)}, \\
&= \frac{\frac{1}{2}n(n-1)(n-1-1)}{\frac{1}{2}\frac{1}{2}n(n-1)\frac{1}{2}(n(n-1) - 2)}, \\
&= \frac{n-2}{\frac{1}{4}(n+1)(n-2)}, \\
&= \frac{4}{n+1},
\end{aligned}
$$

making $\quad \lim_{m \to \infty} \text{density}(H_m) = 0 \Rightarrow \{H_m\}_m \in S.$

2. Suppose $G_n$ is an $r$-regular graph. As $n$ grows each $G_n$ is connected to $r$ nodes. Then deg $v_i = r$ and $m = rn/2$. The ratio $m/n^2 = r/2n$ assigning $\{G_n\}_n \in S$. The density of $H_m$ is given by

$$
\begin{aligned}
\text{density}(H_m) &= \frac{\frac{1}{2}nr^2 - \frac{1}{2}nr}{\frac{1}{2}\frac{1}{2}rn(\frac{1}{2}rn - 1)}, \\
&= \frac{\frac{1}{2}rn(r-1)}{\frac{1}{2}\frac{1}{2}rn\frac{1}{2}(rn - 2)}, \\
&= \frac{4(r-1)}{rn - 2}, \\
&= \frac{2(r-1)}{m - 1}.
\end{aligned}
\tag{6}
$$

Thus, $\lim_{m \to \infty} \text{density}(H_m) = 0$, making both $\{G_n\}_n, \{H_m\}_m \in S$. Using Theorem 3.6 we can conclude $\{G_n\}_n \notin S_q$ because $\{G_n\}_n \in S_q \iff \{H_m\}_m \in D$. Hence $\{G_n\}_n \in S\backslash S_q$. As $G$ is an $r$-regular graph, $H$ is a $2(r-1)$-regular graph with $\frac{nr}{2}$ vertices (Lemma 2.3-2). Thus, using the same reasoning we have $\{H_m\}_m \in S\backslash S_q$. This is an example where both graph sequences $\{G_n\}_n$ and $\{H_m\}_m$ are sparse and both $\{G_n\}_n, \{H_m\}_m \in S\backslash S_q$.

3. Suppose $G_n$ is a path. Then $m = n - 1$ and the starting and ending vertices have degree 1 and the rest have degree 2. Thus,

$$
\begin{aligned}
\text{density}(H_m) &= \frac{\frac{1}{2}(1^2 + (n-2)2^2 + 1^2) - m}{\frac{1}{2}m(m-1)}, \\
&= \frac{\frac{1}{2}(2 + 4(m-1)) - m}{\frac{1}{2}m(m-1)}, \\
&= \frac{1 + 2(m-1) - m}{\frac{1}{2}m(m-1)}, \\
&= \frac{m - 1}{\frac{1}{2}m(m-1)}, \\
&= \frac{2}{m} = \frac{2}{n-1}.
\end{aligned}
$$

Thus, $\lim_{m \to \infty} \text{density}(H_m) = 0$. The edge density can also be derived by recognizing a path of $n$ vertices gives rise to a line graph that is a path of $n - 1$ vertices (Lemma 2.3-3). Using the same reasoning as previously for $r$-regular graphs, we can conclude that both $\{G_n\}_n, \{H_m\}_m \in S\backslash S_q$.

4. Suppose $G_n$ is a cycle, i.e. $G_n = C_n$. Then $n = m$ and all vertices have degree 2. This is a 2-regular graph. Using equation (6) we get

$$\text{density}(H_m) = \frac{4(r-1)}{rn-2},$$

$$= \frac{4}{2n-2} = \frac{2}{n-1} = \frac{2}{m-1},$$

which limits to zero. From Lemma 2.3-5 we know that $L(C_n) = C_n$. Here too both $\{G_n\}_n, \{H_m\}_m \in S \backslash S_q$ as previously.

$\square$

## C  Proofs on results for probabilistic graphs

**Lemma C.1.** *Consider the graph $G_n$ sampled from a $G(n,p)$ model and suppose $G_n$ has $n$ nodes and $m$ edges. Let $X_{ij}$ denote the random variable corresponding to the edge between vertices $i$ and $j$, i.e., $X_{ij} = 1$ if the edge exists and 0 otherwise. Let $Y_j = \sum_i X_{ij}$, $\mu = \mathbb{E}[Y_j]$ and $\bar{m} = \mathbb{E}[m]$. Let $Y_{ssq} = \sum_j Y_j^2$, and $m_{sq} = m^2$. Then for a given $\alpha \in (0,1)$ and $c > 0$ we have*

$$P\left[Y_{ssq} \geq cm_{sq} | m_{sq} \leq (1-\alpha)^2 \bar{m}^2\right] P\left[m_{sq} \leq (1-\alpha)^2 \bar{m}^2\right] \leq \exp\left(-\frac{\alpha^2 pn(n-1)}{4}\right).$$

*Proof.* For a given $\alpha \in (0,1)$ we get the following Chernoff-Hoeffding bounds (Frieze & Karoński, 2015) for $m$:

$$P\left[m \leq (1-\alpha)\bar{m}\right] \leq \exp\left(-\frac{\alpha^2 \bar{m}}{2}\right),$$

$$\text{giving us} \quad P\left[m^2 \leq (1-\alpha)^2 \bar{m}^2\right] \leq \exp\left(-\frac{\alpha^2 \bar{m}}{2}\right),$$

as $m$ is positive. As the probability $P\left[Y_{ssq} \geq cm_{sq} | m_{sq} \leq (1-\alpha)^2 \bar{m}^2\right] \leq 1$, $m_{sq} = m^2$ and $\bar{m} = pn(n-1)/2$ we get the desired result. $\square$

**Lemma C.2.** *Consider the graph $G_n$ sampled from a $G(n,p)$ model and suppose $G_n$ has $n$ nodes and $m$ edges. Let $X_{ij}$ denote the random variable corresponding to the edge between vertices $i$ and $j$, i.e., $X_{ij} = 1$ if the edge exists and 0 otherwise. Let $Y_j = \sum_i X_{ij}$, $\mu = \mathbb{E}[Y_j]$ and $\bar{m} = \mathbb{E}[m]$. Let $Y_{ssq} = \sum_j Y_j^2$, and $m_{sq} = m^2$. Then for a given $\alpha \in (0,1)$ and $c > 0$ we have*

$$P\left[Y_{ssq} \geq cm_{sq} | m_{sq} \geq (1+\alpha)^2 \bar{m}^2\right] P\left[m_{sq} \geq (1+\alpha)^2 \bar{m}^2\right] \leq \exp\left(-\frac{\alpha^2 pn(n-1)}{6}\right).$$

*Proof.* The proof is similar to Lemma C.1 with the only difference being the Chernoff-Hoeffding bound, which changes to:

$$P\left[m \geq (1+\alpha)\bar{m}\right] \leq \exp\left(-\frac{\alpha^2 \bar{m}}{3}\right).$$

$\square$

**Lemma C.3.** *Consider the graph $G_n$ sampled from a $G(n,p)$ model and suppose $G_n$ has $n$ nodes and $m$ edges. Let $X_{ij}$ denote the random variable corresponding to the edge between vertices $i$ and $j$, i.e., $X_{ij} = 1$ if the edge exists and 0 otherwise. Let $Y_j = \sum_i X_{ij}$, $\mu = \mathbb{E}[Y_j]$ and $\bar{m} = \mathbb{E}[m]$. Let $Y_{ssq} = \sum_j Y_j^2$, and $m_{sq} = m^2$. Then for a fixed $c > 0$ and fixed $\alpha \in (0,1)$ for $n > \frac{4}{c(1-\alpha)^2}$ and $\beta = \frac{\sqrt{cn}(1-\alpha)}{2} - 1$ we have*

$$P\left[Y_{ssq} \geq cm_{sq} | (1-\alpha)^2 \bar{m}^2 \leq m_{sq} \leq (1+\alpha)^2 \bar{m}^2\right] P\left[(1-\alpha)^2 \bar{m}^2 \leq m_{sq} \leq (1+\alpha)^2 \bar{m}^2\right]$$

$$\leq \exp\left(\ln n - \frac{\beta^2 p(n-1)}{3}\right).$$

*Proof.* We focus on the term $P\left[Y_{\text{ssq}} \geq cm_{\text{sq}}|(1-\alpha)^2\bar{m}^2 \leq m_{\text{sq}} \leq (1+\alpha)^2\bar{m}^2\right]$. We know that $\mu = p(n-1)$ and $\bar{m} = pn(n-1)/2$. As $m_{\text{sq}} \in \left[(1-\alpha)^2\bar{m}^2, (1+\alpha)^2\bar{m}^2\right]$ we get

$$P\left[Y_{\text{ssq}} \geq cm_{\text{sq}}|(1-\alpha)^2\bar{m}^2 \leq m_{\text{sq}} \leq (1+\alpha)^2\bar{m}^2\right] \leq P\left[Y_{\text{ssq}} \geq c(1-\alpha)^2\bar{m}^2\right],$$

$$= P\left[Y_{\text{ssq}} \geq c\left(\frac{n\mu}{2}\right)^2(1-\alpha)^2\right], \tag{7}$$

$$= P\left[Y_{\text{ssq}} \geq n\mu^2(1+\beta)^2\right], \tag{8}$$

where we have substituted $\bar{m} = \frac{n\mu}{2}$ in equation (7) and rearranged the terms for $\beta$. For $n > \frac{4}{c(1-\alpha)^2}$, we get $\frac{\sqrt{cn}(1-\alpha)}{2} > 1$ making $\beta > 0$.

For a given $\beta > 0$, we get the following Chernoff-Hoeffding bound for $Y_j$:

$$P\left[Y_j \geq (1+\beta)\mu\right] \leq \exp\left(-\frac{\beta^2\mu}{3}\right).$$

As $Y_j \geq 0$ we have $\quad P\left[Y_j^2 \geq (1+\beta)^2\mu^2\right] \leq \exp\left(-\frac{\beta^2\mu}{3}\right),$

and from Boole's inequality $\quad P\left[Y_{\text{ssq}} \geq n(1+\beta)^2\mu^2\right] \leq n\exp\left(-\frac{\beta^2\mu}{3}\right). \tag{9}$

Substituting equation (9) in equation (8) we get

$$P\left[Y_{\text{ssq}} \geq cm_{\text{sq}}|(1-\alpha)^2\bar{m}^2 \leq m_{\text{sq}} \leq (1+\alpha)^2\bar{m}^2\right] \leq n\exp\left(-\frac{\beta^2\mu}{3}\right),$$

$$\leq \exp\left(\ln n - \frac{\beta^2\mu}{3}\right)$$

for $n > \frac{4}{c(1-\alpha)^2}$ and $\beta = \frac{\sqrt{cn}(1-\alpha)}{2} - 1$. $\qquad \square$

**Theorem 5.2.** *Let $G_n$ be an Erdős–Rényi graph sampled from a $G(n,p)$ model and suppose $G_n$ has $n$ nodes and $m$ edges. Let $H_m = L(G_n)$. Then for any $c \in (0,1)$, the edge density of $H_m$ satisfies*

$$P\left[density(H_m) \geq c\right] \leq \exp\left(-\frac{\alpha^2pn(n-1)}{4}\right) + \exp\left(\ln n - \frac{\beta^2p(n-1)}{3}\right) + \exp\left(-\frac{\alpha^2pn(n-1)}{6}\right),$$

*where $\alpha \in (0,1)$, $n > \frac{4}{c(1-\alpha)^2}$ and $\beta = \frac{\sqrt{cn}(1-\alpha)}{2} - 1$. Therefore, as $n$ and $m$ go to infinity the edge density of $H_m$ satisfies*

$$\lim_{m\to\infty} P\left[density(H_m) = 0\right] = 1.$$

*Proof.* Let $X_{ij}$ denote the Bernoulli random variable corresponding to the edge between nodes $i$ and $j$ in $G_n$ and let $X_{ij} = 1$ if the edge is present and 0 otherwise. Let $Y_j = \sum_i X_{ij}$. Then the degree of each node $j$ in $G_n$ is given by $\deg v_j = Y_j$. Let $\mu = \mathbb{E}[Y_j]$ and $\bar{m} = \mathbb{E}[m]$. We know that $\mu = p(n-1)$ and $\bar{m} = pn(n-1)/2$. Let $Y_{\text{ssq}} = \sum_j Y_j^2$, and $m_{\text{sq}} = m^2$ where ssq denotes the sum of squares and sq denotes square.

We fix $c$ and $\alpha$ such that $c, \alpha \in (0,1)$ and compute $P\left[Y_{\text{ssq}} \geq cm_{\text{sq}}\right]$ using the law of total probability

$$P\left[Y_{\text{ssq}} \geq cm_{\text{sq}}\right] = P\left[Y_{\text{ssq}} \geq cm_{\text{sq}}|m_{\text{sq}} \leq (1-\alpha)^2\bar{m}^2\right] P\left[m_{\text{sq}} \leq (1-\alpha)^2\bar{m}^2\right] +$$
$$P\left[Y_{\text{ssq}} \geq cm_{\text{sq}}|(1-\alpha)^2\bar{m}^2 \leq m_{\text{sq}} \leq (1+\alpha)^2\bar{m}^2\right] P\left[(1-\alpha)^2\bar{m}^2 \leq m_{\text{sq}} \leq (1+\alpha)^2\bar{m}^2\right] +$$
$$P\left[Y_{\text{ssq}} \geq cm_{\text{sq}}|m_{\text{sq}} \geq (1+\alpha)^2\bar{m}^2\right] P\left[m_{\text{sq}} \geq (1+\alpha)^2\bar{m}^2\right],$$

$$\leq \exp\left(-\frac{\alpha^2pn(n-1)}{4}\right) + \exp\left(\ln n - \frac{\beta^2p(n-1)}{3}\right) + \exp\left(-\frac{\alpha^2pn(n-1)}{6}\right),$$

from Lemmas C.1, C.2 and C.3 for $n > \frac{4}{c(1-\alpha)^2}$ and $\beta = \frac{\sqrt{cn}(1-\alpha)}{2} - 1$.

For a fixed $c \in (0, 1)$ we get

$$\lim_{n,m\to\infty} P\left[Y_{\mathrm{ssq}} \geq cm_{\mathrm{sq}}\right] = 0$$

As

$$P\left[\frac{1}{2}Y_{\mathrm{ssq}} \geq \frac{1}{2}cm_{\mathrm{sq}}\right] = P\left[\frac{1}{2}\sum_j Y_j^2 - m \geq \frac{1}{2}cm^2 - m\right]$$

For $c \in (0, 1)$ we have $\frac{1}{2}cm^2 - \frac{1}{2}cm > \frac{1}{2}cm^2 - m$ giving us

$$P\left[\frac{1}{2}\sum_j Y_j^2 - m \geq \frac{1}{2}cm^2 - \frac{1}{2}cm\right] \leq P\left[\frac{1}{2}\sum_j Y_j^2 - m \geq \frac{1}{2}cm^2 - m\right] = P\left[Y_{\mathrm{ssq}} \geq cm_{\mathrm{sq}}\right].$$

This gives us

$$P\left[\frac{\frac{1}{2}\sum_j Y_j^2 - m}{\frac{m(m-1)}{2}} \geq c\right] \leq P\left[Y_{\mathrm{ssq}} \geq cm_{\mathrm{sq}}\right],$$

$$P\left[\mathrm{density}(H_m) \geq c\right] \leq \exp\left(-\frac{\alpha^2 pn(n-1)}{4}\right) + \exp\left(\ln n - \frac{\beta^2 p(n-1)}{3}\right) + \exp\left(-\frac{\alpha^2 pn(n-1)}{6}\right),$$

where we have used the line graph edge density in equation (1). As $n$ and $m$ go to infinity

$$\lim_{m\to\infty} P\left[\mathrm{density}(H_m) \geq c\right] = 0,$$

giving us the first result. Taking the complement we have

$$\lim_{m\to\infty} P\left[\mathrm{density}(H_m) < c\right] = 1,$$

for a fixed $c \in (0, 1)$. As this is true for any $c \in (0, 1)$ we have

$$\lim_{m\to\infty} P\left[\mathrm{density}(H_m) = 0\right] = 1.$$

□