# OpenReview forum: "Graphons of Line Graphs"
_TMLR — Rejected by TMLR_

### Review · Reviewer_RjDf · 2025-09-12

**Summary Of Contributions:**

The paper proposes using graphons of line graphs to address challenges in analyzing sparse graph sequences, where standard graphons collapse to zero. The key idea is that applying the line-graph transformation can yield non-zero, informative graphons even when the original graphs are sparse.
The authors begin by reviewing the  line-graph transformation  and the foundations of graphons, including their definition, representation for finite graphs, metrics for comparing them, and conditions for convergence.
Subsequently they show that some sparse graphs in a sequences have dense line graphs if they satisfy the square-degree property, which requires that the sum of squared degrees is proportionally large compared to the square of the total degree sum.
Sequences of sparse graphs that meet this condition, such as star graphs or certain preferential attachment graphs, yield line graphs with non-zero graphons, while dense graphs or sparse graphs without this property always converge to the zero graphon.
Additionally, the resulting graphons from line graphs are orthogonal to the graphons of the original graphs, highlighting that line graphs reveal new structural information not captured by the original sparse graph.
Empirical experiments show that U-random graphs from line-graph graphons better preserve structural properties than W-random graphs of the original sparse graphs.
Finally, the authors show that Superlinear preferential attachment graphs almost surely satisfy the square-degree property, so their line graphs converge to non-zero graphons, capturing the hub-dominated structure. In contrast, Erdős–Rényi graphs produce line graphs whose edge density vanishes, yielding zero graphons despite the original graphs being dense.

**Audience:**

No

**Audience Explanation:**

Despite the paper’s rigor, it is not clearly related to machine learning. While it briefly mentions potential applications of graphons in ML and touches on this in Section 2.3.2, it contains no machine learning theorems, experiments, or demonstrations of practical relevance to the ML community.

**Broader Impact Concerns:**

There is no obvious cause of concern regarding the ethical implications of the work.

**Claims And Evidence:**

Yes

**Claims Explanation:**

Overall, the paper’s claims are well-supported: the theoretical results are backed by lemmas and proofs, and the empirical experiments demonstrate that line-graph graphons effectively distinguish a specific subclass of sparse graph structures.

**Requested Changes:**

1. Run predictive experiments using graphon-based line graphs: Since the introduction mentions that graphons can predict network properties, the authors can use some real network datasets to test whether the structural properties of the predicted and the real network are similar.

2. Apply the methodology to ML tasks: The paper cites graph embedding as potential machine learning applications. They should show how line-graph graphons could be integrated into these tasks—e.g., compare embeddings generated from original sparse graphs versus line-graph graphons in a node classification and link prediction experiments in real and simulated sparse graphs that have the square-degree property.

---

### Review · Reviewer_stUe · 2025-09-17

**Summary Of Contributions:**

This paper introduces a novel approach to studying sparse graph limits through the graphons of their line graphs. The key contribution is identifying the "square-degree property", which characterizes when sparse graphs produce dense line graphs, enabling the application of standard dense graphon theory to a subset of sparse graphs.

Key strengths:
- Elegant and simple idea with a clear structural condition (square-degree property).
- Theoretical contributions are well motivated and supported by proofs.
- Examples (stars, PA graphs, Erdős–Rényi) provide intuitive contrasts.
- Well-structured mathematical exposition.

Key weaknesses:
- Very limited applicability - the square-degree property is satisfied by only a narrow class of sparse graphs.
- Lack of compelling practical applications or use cases.
- Comparison with existing sparse graphon frameworks (e.g., graphexes, stretched graphons) is not deeply discussed.

**Additional Comments:**

The paper makes a valid theoretical contribution but falls short of TMLR's standards for practical relevance. The square-degree property is mathematically elegant but appears too restrictive for real-world applications. I would encourage the authors to expand both the comparative discussion with existing frameworks and the empirical illustrations.

**Audience:**

Yes

**Audience Explanation:**

Researchers working specifically on graph limits and graphon theory would find the mathematical contributions interesting. The novel connection between line graphs and sparse graphons offers a fresh perspective that could inspire further theoretical work. However, the practical machine learning audience would likely find limited value due to the restrictive nature of the square-degree property and lack of demonstrated applications.

**Broader Impact Concerns:**

None identified. This work is primarily theoretical, with no apparent negative ethical implications.

**Claims And Evidence:**

Yes

**Claims Explanation:**

The authors provide clear proofs for their main results, particularly Theorem 3.6. The examples are appropriate for illustrating the central results, but are relatively weak - Section 4.4 only tests star graphs, which are the simplest possible case. The paper would benefit from more diverse experimental validation. Additionally, while the mathematical claims are supported, the practical utility claims in the introduction lack substantive evidence.

**Requested Changes:**

- Add realistic examples of graph families satisfying the square-degree property beyond stars.
- Comparing the line graph graphons to at least one existing sparse graphon framework.
- Include a concrete application example showing how line graph graphons would be used in practice (edge prediction, for example).

- Standardize notation throughout (particularly "deg $v$" vs "deg $v_{i, n}$").

---

### Review · Reviewer_qsCc · 2025-11-12

**Summary Of Contributions:**

The theory of graph limits as originated by Lovasz and Szegedy has lead to many interesting new developments and applications regarding large graphs. One drawback of the theory is that it only gives interesting results when applied to sequences of graphs that are {\sl dense}, i.e.~the number of edges of graphs in the sequences is $\theta (n^2)$, where $n$ is the number of vertices. For sequences of sparse graphs, the theory applies, but the limiting object is the zero graphon. This paper shows that, for some sequences of sparse graphs, graph limit theory can instead be applied to the corresponding sequence of {\sl line graphs}.

The paper gives a simple condition derived from the degree sequence of the graphs, called the square-degree property (Definition 3.3). It follows directly from the definition that dense graph sequences do not satisfy the square-degree condition (Lemma 3.4). The main result of the paper, stated in Theorem 3.6, is that a graph sequence has a dense line graph sequence if and only if it satisfied the square-degree property. The proof is straightforward. It follows from a counting argument and applying the definition.

**Audience:**

Yes

**Audience Explanation:**

The paper starts with an interesting observation, but fails to explore the ramifications of this observation. The paper lacks coherence and direction. It is repetitive and could be greatly condensed. I believe it is of very limited interest in its present form. I do not recommend acceptance for publication.

**Broader Impact Concerns:**

I have no concerns about the ethical implications of this work.

**Claims And Evidence:**

No

**Claims Explanation:**

The main result as described in the previous paragraph is simple but of some interest. It brings up questions and avenues of further study. For example, one might ask whether the graphon that is the limit of a sequence of line graph has any special structure. Can such graphons be recognized by their homomorphism densities? Is it possible to construct or sample graphs from such a graphon that are similar in structure to the sparse graphs in the sequence that gave rise to the line graph sequence? How does the square-degree condition restrict the degree distribution? No such questions are addressed here. The paper gives the trivial example of a collection of distinct stars, whose line graph is a collection of disjoint cliques. In Section 5.1 the authors use a previous result to show that graphs produced by the Preferential Attachment model satisfy the degree-square condition. However, nothing is mentioned about what happens for the Preferential Attachment model in the more familiar and often used linear regime ($\alpha =1$).

In Section 4.4, simulation results are given, which compare the line graphs of graphs sampled from a ${0,1}$-valued graphon representing a graph $G$ to graphs sampled from the graphon  of the line graph of $G$. The options chosen for $G$ are either a star, or two disjoint stars. Note that sampling from a graphon representing a graph $G$ (referred to in this paper as {\sl empirical graphon}) corresponds to a simple graph operation : replace each vertex of $G$ by a clique of approximately equal size, and replace each edge by a complete bipartite graph between those cliques. If $G$ is a star or collection of stars, the effect on edge density and vertex density of this operation can be calculated precisely. The simulation does not give additional insights.

**Requested Changes:**

If editors decide that the main result of the paper is worth publication, then the paper should be rewritten and greatly reduced in length, and its appendices removed. There is a lot of repetition in its current form. Figures such as Figure 3 and 5 do not enlighten but merely restate in a different form what is already mentioned in the text. The lemmas in Section 3.3 and 3.4 are direct corollaries of the main result, to the point that no proof is required. Where this is not exactly the case it can be made so by limiting the discussion to converging graph sequences.

Other examples: Lemma 4.1 establishes that the limiting density of a sequence of star graphs $K_{1,n}$ equals 1. This follows immediately from the fact that the line graph of a star graph is a complete graph. The proof in the appendix is unnecessary. Theorem 5.2 establishes that a.a.s.~line graphs of graphs drawn from the Erd\H{o}s-Renyi graph $G(n,p)$, where $p$ is a constant, are sparse.  The proof, which is based on well-known results regarding the degrees and number of edges of the random graph, can be reduced to a paragraph.

---

### Author Response · Authors · 2025-11-20
**Response to review comments**

We thank the reviewers for their comments and for their time. We acknowledge the limitations of this work. Our aim was to show that line graphs can be used to model sparse graph using graphons. We believe this work provides a different, yet simple avenue for exploring graphon-based approaches to sparse graph modeling and hope it inspires further research in this direction.

---

### Decision · Action_Editor_vsFu · 2026-01-05

**Recommendation:** Reject

**Audience:**

No

**Audience Explanation:**

I think all reviewers agree that the paper agree the paper begins with an interesting observation. However, one reviewer summarized that "the work fails to explore the ramifications of this observation", and they believe it is of limited interest in the present form. Moreover, the current form of the paper has limited applicability to machine learning, both in its presentation and empirical studies. All reviewers gave more suggestions on how to improve the manuscript for the ML and graphs audience.

**Claims And Evidence:**

No

**Claims Explanation:**

This paper focuses on the theory of graph limits as applied to the setting of line graphs. The paper formalizes the square-degree property, and shows this condition characterizes the denseness of the induced line graph. Reviewers appreciated this observation, remarking that it is elegant and interesting.

From the initial reviewers, all reviewers pointed out that the paper is of limited depth and interest to the TMLR community. The main reasons for rejection are due to:
* Limited exploration of the consequences of the ramifications of the square-degree property. One reviewer pointed out that central questions of interest are not adequately explored: "One might ask whether the graphon that is the limit of a sequence of line graph has any special structure. Can such graphons be recognized by their homomorphism densities? Is it possible to construct or sample graphs from such a graphon that are similar in structure to the sparse graphs in the sequence that gave rise to the line graph sequence? How does the square-degree condition restrict the degree distribution?"
* Writing: The paper is repetitive and could be presented more concisely. The reviews provide suggestions on how to improve.
* Reviewers requested additional discussion that motivates the square-degree property and its connection to real-world applications, both discussion it more comparatively with existing framework and also in the empirical illustrations.
* The practical relevance to machine learning is limited in terms of its theory, experiments, or demonstrations.

Finally, no author response to these comments or revision was submitted.

We encourage the authors to revise the manuscript according to the reviewer feedback.